# Prognostic MicroRNA Panel for HCV-Associated HCC: Integrating Computational Biology and Clinical Validation

**DOI:** 10.3390/cancers14133036

**Published:** 2022-06-21

**Authors:** Areeg M. Dabbish, Hana M. Abdelzaher, Moustafa Abohawya, Samir Shamma, Yosra H. Mahmoud, Amr Maged, Mohamed Manaa, Mohamed Hassany, Firas Kobeissy, Omid Bazgir, Hassan El-Fawal, Hassan M. E. Azzazy, Anwar Abdelnaser

**Affiliations:** 1Biotechnology Graduate Program, Department of Biology, School of Sciences and Engineering, The American University in Cairo, New Cairo 11835, Egypt; adabbish@aucegypt.edu (A.M.D.); moustafa_abohawya@aucegypt.edu (M.A.); 2Institute of Global Health and Human Ecology (IGHHE), The American University in Cairo, New Cairo 11835, Egypt; hana-abdelzaher@aucegypt.edu (H.M.A.); samirnabhan@aucegypt.edu (S.S.); hassan.elfawal@aucegypt.edu (H.E.-F.); 3Fellow of Clinical Pathology, National Hepatology and Tropical Medicine Research Institute (NHTMRI), Cairo 11562, Egypt; yosrahosny34@gmail.com; 4Tropical Medicine Department, National Hepatology and Tropical Medicine Research Institute (NHTMRI), Cairo 11562, Egypt; amrrasheed88@gmail.com (A.M.); dr_mo_mana3@hotmail.com (M.M.); mohamed.hassany@mohealth.gov.eg (M.H.); 5Program for Neurotrauma, Neuroproteomics & Biomarkers Research, Departments of Emergency Medicine, Psychiatry, Neuroscience and Chemistry, University of Florida, Gainesville, FL 32611, USA; firasko@gmail.com; 6Department of Biochemistry and Molecular Biology, American University of Beirut, Beirut 11-0236, Lebanon; 7Modeling and Simulation/Clinical Pharmacology, Genentech, CA 94080, USA; omidbazgir00@gmail.com; 8Department of Chemistry, The American University in Cairo, New Cairo 11835, Egypt; hazzazy@aucegypt.edu

**Keywords:** hepatocellular carcinoma, hepatitis C virus, microRNAs, serum biomarker, HCC prognosis

## Abstract

**Simple Summary:**

Hepatocellular carcinoma (HCC) is a disease of poor prognosis. The early diagnosis of HCC will restrain the disease progression and thus improve patients’ quality of life. We aimed in this study to define a panel of microRNAs (miRNAs) that could facilitate the early prognosis of HCC focal lesions in the cirrhotic livers of patients previously infected with hepatitis C virus (HCV). A minimally invasive technique was used to measure the differential expression of isolated miRNAs from the serum of 201 HCV infected and HCV-HCC patients. We suggest that a panel of five serum miRNAs (miR-150, miR-199a, miR-224, miR-424, and miR-3607) might provide a potentially promising tool in the prognosis of HCC disease in cirrhotic HCV patients.

**Abstract:**

Early detection of hepatocellular carcinoma (HCC) will reduce morbidity and mortality rates of this widely spread disease. Dysregulation in microRNA (miRNA) expression is associated with HCC progression. The objective is to identify a panel of differentially expressed miRNAs (DE-miRNAs) to enhance HCC early prediction in hepatitis C virus (HCV) infected patients. Candidate miRNAs were selected using a bioinformatic analysis of microarray and RNA-sequencing datasets, resulting in nine DE-miRNAs (miR-142, miR-150, miR-183, miR-199a, miR-215, miR-217, miR-224, miR-424, and miR-3607). Their expressions were validated in the serum of 44 healthy individuals, 62 non-cirrhotic HCV patients, 67 cirrhotic-HCV, and 72 HCV-associated-HCC patients using real-time PCR (qPCR). There was a significant increase in serum concentrations of the nine-candidate miRNAs in HCC and HCV patients relative to healthy individuals. MiR-424, miR-199a, miR-142, and miR-224 expressions were significantly altered in HCC compared to non-cirrhotic patients. A panel of five miRNAs improved sensitivity and specificity of HCC detection to 100% and 95.12% relative to healthy controls. Distinguishing HCC from HCV-treated patients was achieved by 70.8% sensitivity and 61.9% specificity using the combined panel, compared to alpha-fetoprotein (51.4% sensitivity and 60.67% specificity). These preliminary data show that the novel miRNAs panel (miR-150, miR-199a, miR-224, miR-424, and miR-3607) could serve as a potential non-invasive biomarker for HCC early prediction in chronic HCV patients. Further prospective studies on a larger cohort of patients should be conducted to assess the potential prognostic ability of the miRNAs panel.

## 1. Introduction

Hepatocellular carcinoma (HCC) is the most predominant type of primary liver cancer; it encompasses 75–85% of liver cancer cases [1]. It is the sixth most prevalent cause of cancer [2] and the third leading cause of cancer mortality worldwide [3]. An estimate of 905,577 (4.7%) newly diagnosed HCC patients and around 830,130 death cases (8.3%) were globally recorded in 2020 [4,5]. Egypt was ranked the third in Africa and the fifteenth globally in HCC prevalence, and the documented cases were doubled over a decade, resulting in more challenging health problems [6]. One of the most predominant risk factors for HCC development and progression is chronic hepatitis C virus (HCV) infection. Egypt has an exceptionally high prevalence of HCV worldwide [7]. According to the Egyptian demographic health survey in 2015, almost 6 million Egyptians aged 15–59 were chronically infected with HCV. Extensive research and massive efforts are exerted to overcome the viral spread through national eradication programs and explore novel anti-HCV therapies.

Diagnosis of HCC remarkably changed over the previous decade to switch from invasive techniques such as angiography and tissue biopsy to non-invasive imaging procedures including ultrasound (US), computed tomography (CT), and magnetic resonance imaging (MRI), in addition to serological testing using alpha-fetoprotein (AFP) [8]. Although AFP is considered the most extensively used serum biomarker for HCC, it is characterized by low sensitivity and specificity in determining HCC lesions even at low-level cutoffs (10–20 ng/mL) [9]. It has 25% sensitivity for the detection of small tumors (less than 3 cm), and the sensitivity could reach 50% for focal lesions (FL) larger than 3 cm [10]. The AFP level increases in some benign liver diseases such as liver cirrhosis (LC) and hepatitis [11], and a normal serum AFP level is detected in 15–30% of advanced HCC cases [12]. Therefore, the Practice Guideline Committee of the American Association for the Study of Liver Disease (AASLD) does not recommend using AFP in the early detection of HCC [13]. There is no sole serum biomarker, including immunoglobulin M (IgM) immunocomplexes, that could provide an accurate diagnosis for HCC. Combining multiple biomarkers could improve the efficiency and the sensitivity of the detection [14].

MicroRNAs (miRNAs) are single-stranded, short (approximately 22 nucleotides), non-coding RNAs associated with the argonaute family protein. MiRNAs regulate gene expression through post-transcriptional gene silencing [15]. Generally, most protein-coding genes are influenced by miRNAs [16]; thus, deregulation in miRNAs’ expression is directly correlated with many physiological disorders such as cancer [17]. Numerous profiling studies address miRNAs’ expression in HCC, and marked changes in miRNAs’ expression were recorded in human HCC tissues compared to non-tumorous liver tissues [18]. Therefore, it is suggested that disordered miRNAs’ biogenesis further triggers the current miRNAs’ deregulation to enhance HCC and metastasis [19,20]. The non-coding RNA expression levels are correlated with tumor size, tumor stage, cirrhotic state, and patients’ overall survival [19,20,21,22,23]. The abundance of stable miRNAs in the circulation (serum and plasma) of healthy individuals and HCC patients with different expression patterns suggested a promising role for these miRNAs in the prognosis of the liver carcinogenesis [24,25].

The current research hypothesized that computationally assigned miRNAs possess differential expression between HCC patients and HCV-infected subjects. The miRNAs panel can predict HCC in chronic HCV patients using minimally invasive serum samples. Thus, the current study suggested a novel miRNAs panel that potentially acts as a prognostic biomarker for HCC early detection.

## 2. Materials and Methods

### 2.1. Bioinformatic Analysis

To identify the target miRNAs panel, bioinformatic analysis was performed on a non-coding RNA microarray dataset (GSE40744) deposited on the National Center for Biotechnology Information—Gene expression omnibus (NCBI-GEO) repository [26]. In such a study, Diaz et al. performed microarray analysis for miRNAs in HCV-associated HCC in addition to other liver conditions of healthy control samples. They used 26 liver specimens isolated from 10 HCV-HCC patients, divided as nine specimens from the HCC FLs and 17 specimens from the surrounding non-tumorous tissues with a cirrhotic nature [26]. Moreover, they included 18 specimens from 10 HCV- cirrhotic patients, 13 specimens from 4 patients diagnosed with HBV-associated acute liver failure, 12 specimens from liver donors, and 7 normal liver specimens from subjects who performed hepatic resection liver angioma [26]. In our study, bioinformatics analysis was performed using 18 specimens of HCV-associated cirrhosis without HCC and 9 specimens from the tumor area of HCV-HCC samples. MiRNA expression profiling was analyzed using R software (R × 64 v.3.6.2). To make this comparison, the R code generated from the GEO2R tool was used to determine differentially expressed miRNAs (DE miRNAs) between these two defined groups of HCV patients. This code used the limma package’s function eBayes to generate a linear model fit and differential expression statistics. The Limma’s function toptable was used later to filter for adjusted *p*-value < 0.01, and the log_2_foldchange 1.5. ggplot2 package was used to generate the volcano plot in Figure 1. Detailed code of data retrieval and analysis is supplemented as Appendix A.

Further analysis of RNA sequencing data deposited on the cancer genome atlas (TCGA) was performed [27]. Data downloaded from the TGCA portal represent miRNA expression and clinical data in HCC patients from different etiologies using the TCGA biolinks GDCquery function with the following arguments: project = “TCGA-LIHC,” data.category = “Transcriptome Profiling” and experimental.strategy = “miRNA-Seq” [28]. HCC expression data with only the HCV etiology were filtered in our study using R software to obtain a smaller dataset composed of 31 samples and were further processed in a downstream analysis. Filtering criteria were applied to retain the miRNAs with an expression of more than ten reads per million in ≥85% of the dataset. After that, differential expression between HCV non-tumor tissue samples and HCV-HCC tissue samples was performed using the TCGABiolinks R package, TCGAanalyze_DEA function. DE miRNAs were defined as having FDR less than 0.05 and log_2_foldchange more than 1. The ggplot2 package was used to plot a volcano plot Figure 2. Detailed code for data retrieval, analysis, and filtering is attached as Appendix A.

### 2.2. Patients and Samples

This study included 245 individuals, 44 healthy volunteers, 62 HCV non-cirrhotic, 67 HCV cirrhotic, and 72 HCC post-HCV infected patients. The three diseased groups include patients chronically infected with HCV, without receiving any treatment for HCV (HCV treatment naïve) or HCV treated patients, who have achieved a sustained virological response (HCV-SVR), which is guaranteed by the absence of HCV particles in the patient’s blood 12 weeks after cessation of HCV therapy. Samples were collected from the National Hepatology and Tropical Medicine Research Institute (NHTMRI) from July to December 2019. Patients’ health history records were compiled, with complete clinical and ultrasonographic examinations.

#### 2.2.1. Inclusion and Exclusion Criteria

Healthy individuals with normal liver functions, no history of viral hepatitis or any liver disease, and a generally good health condition with no major disorders in the kidney, heart, lungs, or other vital organs were included in the study. HCV patients with positive circulating anti-HCV antibodies were classified into two sub-groups based on the presence or the absence of cirrhosis. Diagnosis of features of cirrhosis was made based upon ultrasonography and blood examination (complete blood count, liver function tests as AST, ALT, ALB, T. Bil, and D. Bil). Degree of fibrosis and severity of the liver condition in chronic hepatitis C (CHC) patients was assessed after ultrasonographic examination, which was followed by the application of non-invasive testing as the AST to platelets ratio index (APRI), fibrosis-4 (FIB-4), AST to ALT ratio (AAR) indices, and Child-Turcotte-Pugh (CTP) score for further evaluation. However, for HCV-HCC patients, diagnosis primarily relied on abdominal ultrasonography and triphasic CT scan or MRI to examine the FL and AFP blood levels. The BCLC staging system was used to classify HCC patients based on tumor stage, cancer-related symptoms, and serological liver function tests [29]. HCC patients enrolled in the study were classified into BCLC stages 0, A, and B.

HCV patients who had other viral (e.g., hepatitis B virus) or non-viral liver disease (e.g., alcoholic liver disease or non-alcoholic fatty liver) in conjunction with HCV were excluded from the study. HCC patients with other liver disorders such as hemangioma or cholangiocarcinoma were disqualified. In addition to excluding HCC patients with extrahepatic metastatic cancer, patients with another type of cancer or other comorbid condition such as kidney or heart disorders were also excluded.

#### 2.2.2. Sampling and Serum Preparation

Five mL of blood were withdrawn from each patient into a labeled disposable serum collection tube (global roll gel and clot activator tube). For complete clotting, blood samples were kept for one hour at room temperature (15–25 °C); then, samples were processed for serum separation following the miRNeasy serum/plasma handbook-Qiagen, Hilden, Germany, 2012.

#### 2.2.3. Liver Function Testing and HCV Antibodies Testing

All samples were subjected to qualitative detection of HCV antibodies in the serum using the HCV rapid test cassette (ACON, San Diego, CA, USA—Cat. No. L031-10341). Quantitative determination of liver function biomarkers was assessed for all samples. ALT, ASL, and albumin were measured using the SPINREACT kit (SPINREACT, Barcelona, Spain - Cat. No. 41283, 41273, 1001022, respectively) following the manufacturer’s protocols, while the determination of total serum and direct bilirubin was conducted using the RANDOX kit (County Antrim, UK—Cat. No. BR 412) following the manufacturer’s protocol. The quantitative determination of AFP was conducted for diseased samples only on a fully automated Cobas analyzer using Elecsys AFP (Roche, Mississauga, Canada—Cat. No. 04481798-190).

### 2.3. RNA Isolation and cDNA Synthesis

Total RNA was isolated using the miRNeasy mini kit (Cat. No. 217004—Qiagen, Germany) following the manufacturer’s protocol miRNeasy serum/plasma handbook (February 2012) with minor modifications. Two hundred μL of serum were used to isolate RNA. At the pre-last step, RNeasy MinElute spin columns were left with the lids opened for five min on the benchtop to air dry instead of centrifugation; then, total RNA was eluted in 14 μL RNase free water. Reverse transcription (RT) was performed using the miScript II RT kit (Qiagen, Germany—Cat. No. 218161), following the manufacturer’s miScript PCR system handbook protocol using the RNA volume equivalent to 50 ng.

### 2.4. Real Time PCR Amplification of miRNAs

qPCR of the target miRNAs was performed on the applied biosystems 7500 real time PCR machine (Themo Fisher Scientific, Foster city, CA, USA) using the miScript SYBR Green PCR kit (Qiagen, Germany—Cat. No. 218073) and miScript primer assays (SNORD 68 as a housekeeping gene and specific primers for the target miRNAs—Qiagen, Germany—Cat. No. 218300), following the manufacturer’s protocol of the miScript PCR system handbook, using 1μL cDNA (complementary deoxyribonucleic acid) (concentration: 2.5 ng/μL) in 10 μL total reaction volume. Primers’ sequences are as the following:

hsa-miR-142-3p; 5′UGUAGUGUUUCCUACUUUAUGGA3′,hsa-miR-150-5p; 5′UCUCCCAACCCUUGUACCAGUG3′,hsa-miR-183-5p; 5′UAUGGCACUGGUAGAAUUCACU3′,hsa-miR-199a-3p; 3′ACAGUAGUCUGCACAUUGGUUA5′,hsa-miR-215-5p; 5′AUGACCUAUGAAUUGACAGAC3′,hsa-miR-217-5p, 5′UACUGCAUCAGGAACUGAUUGGA3′,hsa-miR-224-5p; 5′UCAAGUCACUAGUGGUUCCGUUUAG3′,hsa-miR-424-5p; 5′CAGCAGCAAUUCAUGUUUUGAA3′ andhsa-miR-3607-5p; 5′GCAUGUGAUGAAGCAAAUCAGU3′.

Each experiment was performed using two technical replicates. The sequences of miR-199a-3p and 199b-3p on miRbase were similar, and the miR-199a-3p primer was further used in qPCR amplification.

### 2.5. Data Analysis

Fold change of expression was calculated using the formula 2^−∆∆Ct^ [30]. Handling of non-detects (undetermined values) in qPCR results was conducted by excluding all samples that failed to have a true amplification curve. However, if the true amplification curve was recorded, non-detects were replaced by the maximum possible Cycle threshold (Ct) value (Ct = 40). Similarly, Ct values ≥ 40 were replaced by Ct = 40 [31,32,33]. The receiver operator characteristic (ROC) curve was constructed to identify the optimal cut-off value, which maximizes the sum of sensitivity and specificity of each biomarker. The area under the curve (AUC) was calculated. The performances of the individual targets and the combined panels were evaluated by comparing the predicted outcome values (probability of true positive and false positive) with the true outcome variables and calculating the sensitivity and specificity. The optimal cut-off value, sensitivity, and specificity were determined by calculating the Youden index.

### 2.6. Logistic Regression Analysis

A combined logistic regression analysis was conducted to construct the combined panel to validate the miRNAs’ predicted diagnostic significance. All miRNAs were considered as the input of the logistic regression model. The permutation feature importance measure, SHAP (SHapley Additive exPlanations) analysis, and logistic regression weights extraction were utilized to quantify each miRNA’s importance in predicting each patient’s status [34,35]. ROC curves were constructed, and the AUC was determined. The optimal cut-off values were extracted. The sensitivity and specificity of the miRNAs’ prognostic value were calculated to predict the clinical outcome of the combined panel.

### 2.7. Pathway Analysis

To assess the altered pathway analysis and gene interaction networks, we utilized the Elsevier’s PathwayStudio v. 10 [36] (https://www.elsevier.com/solutions/pathway-studio-biological-research, accessed on 9 January 2022) as a dedicated powerful bioinformatics platform. Our microRNA “omics” data were interrogated for functional analysis and interactome assessment. Differential miRNAs were assessed against the propriety PathwayStudio ResNet database gene analysis to achieve this purpose. Furthermore, the “Subnetwork Enrichment Analysis” (SNEA) algorithm, which utilizes Fisher’s statistical test, was used to associate one differential hit (gene/biological process/microRNA) to a specific molecular function, pathway, enabling cellular localization of these gene/pathways. In addition, we performed a targeted analysis that evaluated the direct and non-direct interactome of the altered described miRNAs in relation to HCC.

### 2.8. Statistics

Statistical analysis was conducted using IBM SPSS statistical data editor version 25 (SPSS Inc., Chicago, IL, USA). MDCalc Medical calculator software (version 15.0 for Microsoft Windows, Ostend, Belgium) was used to calculate fibrosis scoring indices. Analysis was conducted using Mann–Whitney U and Kruskal–Wallis H for non-normally distributed data and ANOVA (analysis of variance) for normally distributed quantitative data. Normality testing was performed using the Anderson–Darling and Kolmogorov–Smirnov tests using GraphPad Prism (version 8.4.3). However, analysis of qualitative data was performed using the Chi-square test. Spearman’s rank correlation was used to study the correlation between studied miRNAs in different categories. Figures were designed using SPSS and GraphPad prism. For microarray dataset analysis, the *p*-value was corrected using the Bonferroni method. All statistical tests were two-tailed, and a *p*-value ≤ 0.05 was considered statistically significant.

## 3. Results

### 3.1. Microarray Bioinformatic Analysis

Analysis for microarray datasets was performed using R software. The filtering criterion was set to a log fold change cutoff of 1.5 and an adjusted *p*-value of 0.01 (Figure 1). Twenty-two differentially expressed miRNAs (DE-miRNAs) between cirrhotic HCV and HCC groups were generated. Four miRNAs were upregulated, and 18 were downregulated in the HCC group (log fold change and statistical significance are presented in (Appendix A).

### 3.2. RNA Sequencing Bioinformatic Analysis

MiRNAs’ differential expression analysis was conducted using a false discovery rate (FDR) cutoff of 0.05 and a fold change cutoff of 1 (Figure 2). Nine significant DE-miRNAs were obtained; three were upregulated (miR-217, miR-224, and miR-183); and six were downregulated (miR-142, miR-199b, miR-150, miR-424, miR-215, and miR-3607) (log fold change and FDR are shown in Appendix A).

### 3.3. Description of the Study Population

Demographic and clinical data records are summarized in Table 1 and Figure 3. No significant difference was observed in the gender distribution between the healthy control and HCC group (*p* = 0.5294) and between non-cirrhotic and cirrhotic groups (*p* = 0.1508). Moreover, no statistical difference was observed in the age distribution among the healthy individuals, cirrhotic group, and HCC groups, excluding the non-cirrhotic group (*p* = 0.0662). Differences in serum levels of liver biomarkers—albumin (ALB), alanine aminotransferase (ALT), aspartate aminotransferase (AST), total bilirubin (T. Bil), direct bilirubin (D. Bil), and AFP—were statistically significant among the study groups.

Furthermore, the comparison between HCV cirrhotic and HCV-HCC groups showed no significant difference in the age distribution (*p* = 0.388). In contrast, a significant difference was recorded in the HCV treatment states (*p* = 0.005) and the gender distribution (*p* < 0.001) between the two groups because HCC is a predominantly male disease. The serum levels of some liver function parameters (ALT, T. Bil, D. Bil, and ALB) did not demonstrate any statistical significance between cirrhotic and HCC patients; however, AST and AFP serum levels were significantly altered between the two study groups (*p* = 0.046, <0.0001, respectively). Moreover, the assessment of blood cells’ count (Hemoglobin, RBCs, and WBCs) displayed the statistical difference in HCV cirrhotic compared to the HCC group; only the platelets’ count was not significantly altered between both groups (*p* = 0.481).

Different indices as APRI, FIB-4, AAR were used to assess the degree of fibrosis and cirrhosis in the study groups (Table 2 and Figure 4). Significant fibrosis was calculated in 2.1%, 16.9%, and 23.1% of non-cirrhotic, cirrhotic, and HCC patients, respectively, using the APRI cutoff score > 1.5. However, applying the FIB-4 cutoff score > 3.27 predicated higher percentages of advanced fibrosis in 10.8%, 36%, and 49.2% of non-cirrhotic, cirrhotic, and HCC patients, respectively. Moreover, AAR > 1 suggested the presence of cirrhosis in 75.4%, 73.8%, and 84.3% of the three diseased groups, respectively. Results obtained from the non-invasive indices indicated the importance of combining different indexes and imaging techniques as the US for precise determination of the degree of fibrosis and cirrhosis as an alternative to the transient elastography liver stiffness test.

Tumor features and classification of patients among the different disease stages were assessed (Table 3). Patients were classified into Child A, B, and C in 69.7%, 24.2%, and 6.1% of the cirrhotic patients, in addition to 78.3%, 20.3%, and 1.4% of the HCC patients, respectively, according to the CTP score, while according to Barcelona clinic liver cancer (BCLC) staging systems, the patients were classified into three groups: very early (stage 0; 17.9%), early (stage A; 64.2%), and intermediate stage (stage B; 17.9%). Furthermore, marked slight to moderate ascites were reported in 22.4% of the cirrhotic patients and 22.2% of the HCC patients. Single FL was diagnosed in 63.9% of the HCC patients, while 36.1% had multiple FLs. Additionally, 40.7% of the HCC patients have FL ≤ 3 cm in diameter.

### 3.4. MiRNAs Serum Signature in the Study Groups

Fold changes of the DE-miRNAs among the study groups are represented in Figure 5. The results of this study revealed that serum levels of the nine candidate miRNAs were differentially expressed in HCV and HCC patients in comparison to healthy individuals with high statistical significance (*p*-value < 0.0001) using the Mann–Whitney U statistical test (Table 4). However, only miR-424 serum levels showed statistical significance upon comparing HCV patients with those having HCC. The expression levels of miR-424, miR-199a, miR-142, and miR-224 were significantly altered in HCC patients compared to the non-cirrhotic subjects (*p* < 0.0001, *p* = 0.0001, *p* = 0.023, and *p* = 0.027, respectively), whereas miR-199a and miR-183 showed differences in the differential expression between HCV cirrhotic and non-cirrhotic patients (*p* = 0.012 and *p* = 0.036, respectively) (Table 5).

### 3.5. Correlation between the Studied miRNAs

Spearman’s correlation test was performed to investigate the correlation between the fold change of expression of each individual miRNA and the other miRNAs. A positive correlation was recorded between the expression of all miRNAs among the study groups with high statistical significance (*p* < 0.0001) (Table 6). The highest correlations were recorded between miR-424 and miR-199a (Spearman’s rank correlation coefficient “rho” = 0.828, *p* < 0.0001), miR-424 and miR-150 (rho = 0.774, *p* < 0.0001), miR-424 and miR-224 (rho = 0.769, *p* < 0.0001), miR-224 and miR-150 (rho = 0.767, *p* < 0.0001), miR-224 and miR-199a (rho = 0.727, *p* < 0.0001), miR-183 and miR-217 (rho = 0.725, *p* < 0.0001), and miR-183 and miR-215 (rho = 0.707, *p* < 0.0001). Moreover, the correlation between the miRNAs under study and some clinicopathological characteristics was performed. MiR-183 (rho = 0.206, *p* = 0.007), miR-199a (rho = 0.16, *p* = 0.037), and miR-215 (rho = 0.17, *p* = 0.026) were positively correlated with the age of the patients. While only miR-142 (rho = 0.153, *p* = 0.036) and miR-217 (rho = 0.251, *p*= 0.001) were positively correlated with gender, the whole panel of the nine miRNAs was positively correlated with the cirrhotic liver conditions and D. Bil (*p* < 0.0001), ALT, and AST levels (*p* < 0.001). On the other hand, only miR-150 showed a statistically significant negative correlation with T. Bil levels (rho = −0.219, *p* = 0.003) and with ALB levels for most of the targets (*p* < 0.001). Moreover, the association between the tumor characteristics and the miRNAs’ expression was investigated. Only miR-199a showed a significant positive correlation with the AFP levels (rho = 0.17, *p* = 0.04). For the CTP score, miR-150 was negatively correlated (rho = −0.259, *p* = 0.009), while miR-217 and miR-3607 were positively correlated (rho = 0.231, *p* = 0.02 and rho = 0.214, *p* = 0.032, respectively). Interestingly, no significant correlation was reported with BCLC staging.

Furthermore, a partial correlation analysis was performed to examine the correlation between the fold change of the expression of each individual miRNA and the other miRNAs in the HCV cirrhotic group in comparison to the HCV-HCC group (Appendix A). The highest significant positive correlation was reported between miR-150 and with each of miR-199a, miR-224, miR-424 (rho = 0.386, 0.461, and 0.544, respectively, *p* < 0.0001). Moreover, a highly significant positive correlation was found between miR-424 and each of miR-199a, miR-183, and miR-224 (rho = 0.555, 0.364, and 0.359, respectively, *p* < 0.0001), and between miR-215 with miR-217 (0.529, *p* < 0.0001), in addition to the correlation between miR-199a and miR224 (rho = 0.515, *p* < 0.0001), and between miR-183 and miR-215 (rho = 0.361, *p* < 0.0001). Moreover, miR-217 showed a positive correlation with miR-183 (rho 0.289, *p* = 0.001) and miR-3607 (rho 0.296, *p* = 0.001), 2hile miR-199a had a positive correlation with multiple targets including miR-217 (rho = 0.198, *p* = 0.028), miR-142 (rho = 0.223, *p* = 0.013), miR-215 (rho = 0.246, *p* = 0.006), and miR-183 (rho 0.271, *p* = 0.002). MiR-150 was reported to have a positive correlation with miR-142 (rho = 0.197, *p* = 0.028), while miR-183 was positively correlated with miR-3604 (rho = 0.218, *p* = 0.015). However, unlike the Spearman correlation among the four study groups, the cirrhotic and HCC groups did not show any significant correlation with other demographic (age and gender) or clinical (liver biomarkers and AFP) factors.

### 3.6. Diagnostic Potential of the DE-miRNAs

#### 3.6.1. Diagnostic Potential of the DE-miRNAs in HCC Patients Compared to Healthy Individuals

ROC curves were plotted for the candidate miRNAs to discriminate HCC patients from healthy controls (Figure 6 and Table 7). The AUC values were 0.993, 0.972, 0.968, 0.958, 0.957, 0.933, 0.928, 0.921, 0.868 corresponding to miR-424, miR-142, miR-199a, miR-215, miR-183, miR-217, miR-150, miR-224, and miR-3607, respectively, with high statistical significance (*p*-value < 0.0001). All of the targets showed high sensitivity (ranging from 100% to 80.77%) and accuracy (ranging from 95.7% to 81.72%) for the prediction of HCC patients. A combined panel of five miRNAs, where the whole panel is considered positive if five out of the nine miRNAs tested positive, increased the sensitivity, specificity, and accuracy of detection to 100%, 95.12%, and 97.85%, respectively (*p*-value < 0.0001).

#### 3.6.2. Diagnostic Potential of the DE-miRNAs in HCV Patients Compared to Healthy Individuals

To identify HCV patients from healthy individuals, ROC curves were drawn for the candidate miRNAs (Figure 7 and Table 7). The AUC values were 0.941, 0.94, 0.927, 0.919, 0.913, 0.903, 0.882, 0.863, and 0.824 corresponding to miR-183, miR-424, miR-217, miR-199a, miR-215, miR-142, miR-150, miR-224, and miR-3607, respectively, with high statistical significance (*p*-value < 0.0001). All of the targets showed high sensitivity (ranging from 91.58% to 80%) and accuracy (ranging from 90.44% to 79.41%) for discrimination of HCV patients. A combined panel of five miRNAs improved overall sensitivity, specificity and accuracy of detection to 90.53%, 85.37%, and 88.97%, respectively (*p*-value < 0.0001).

#### 3.6.3. Diagnostic Potential of the DE-miRNAs in HCC Patients Compared to Non-HCC Individuals

In a comparison between HCC patients with others without malignancy (healthy individuals, HCV non-cirrhotic, and HCV cirrhotic patients), AUC values were calculated, and eight potential miRNAs had statistically significant values (Figure 8 and Table 7). MiR-424, miR-199a, miR-150, miR-215, miR-224, miR-142, miR-183, and miR-3607 had AUC values of 761, 0.724, 0.706, 0.695, 0.691, 0.69, 0.664, and 0.664, respectively (*p*-value < 0.0001). Reasonable sensitivities (80.77% to 61.54%) and accuracies (65.96% to 57.98%) were recorded for the different targets. Using a combined panel of five miRNAs resulted in 80.77% sensitivity and 61.03% specificity for HCC detection (*p*-value < 0.0001).

#### 3.6.4. Diagnostic Potential of the DE-miRNAs in HCC (Sustained Virological Response (SVR)/Treatment Naïve) Patients Compared to Non-HCC (SVR/Treatment Naïve) Patients

The overall ROC analysis measurements, including AUC, were improved after classifying the study groups into SVR (Figure 9 and Table 7) treatment naïve (Figure 10 and Table 7). The highest AUC for both the HCC (SVR) group and HCC (treatment naïve) group was recorded for miR-424 (0.8, and 0.835, respectively) (*p*-value < 0.0001). Similarly, the 5-miRNAs combined panel increased the overall sensitivity, specificity, and accuracy for HCC (SVR) patients’ prognosis to 83.33%, 63.73%, and 67.46%, respectively, while combining six miRNAs in one panel improved the calculated measurements in HCC (treatment naïve patients) (sensitivity 89.29%, specificity 72.6%, and accuracy 77.23%).

#### 3.6.5. Diagnostic Potential of the DE-miRNAs in HCC Patients Compared to Patients with HCV

ROC analysis was performed to identify and validate a potential biomarker(s) to differentiate HCC patients from HCV subjects and resulted in three statistically significant targets; miR-424, miR-199a, and miR-150, with *p*-values 0.001, 0.018, and 0.028, respectively (Figure 11 and Table 7). The highest calculations were obtained for miR-424. A comparison between miRNA-424 and AFP (the current HCC serum biomarker) resulted in comparable sensitivities, 63.46% for the former and 62.32% for the latter, although AFP specificity and accuracy (64.57% and 63.78%) were better than those for miR-424 (57.9% and 59.86%). However, the choice of a combined panel of two miRNAs with or without AFP did not provide significant improvement in the ROC analysis measurements.

#### 3.6.6. Diagnostic Potential of the DE-miRNAs in HCC (SVR) Patients Compared to Patients with HCV (SVR)

Similarly, ROC curves were constructed to determine the best AUC for patients with HCC (SVR) compared to HCV (SVR) (Figure 12 and Table 7). Three miRNAs had statistically significant results (miR-424, miR142, and miR-3607) with *p*-values 0.01, 0.018, and 0.014, respectively. The ROC measurements obtained (sensitivities: 66.67%, 66.67%, and 70.83%; accuracies: 62.1%, 68.97%, 58.62%, respectively) were remarkably higher than sensitivity and accuracy recorded for AFP (51.43% and 58.1%, respectively). Using a combined panel of two miRNAs enhanced the sensitivity, specificity, and detection accuracy (70.83%, 61.9%, and 64.37%, respectively). The addition of an AFP biomarker to the combined panel improved the overall sensitivity (70.83%), specificity (73%), and accuracy (72.4%). However, the ROC analysis to discriminate patients with HCC (treatment naïve) from HCV (treatment naïve) patients did not show any statistically significant AUC values for any of the targets.

The ROC curve corresponding to the combined panel in each scenario were constructed and the AUC was calculated to assess the diagnostic potentials of the combined panels (Figure 13).

### 3.7. Establishment of the miRNAs’ Predictive Ability Using Logistic Regression Analysis

For an alternative method for demonstrating the biomarkers with the most discriminatory power, three logistic regression analyses (permutation feature importance measure, SHAP analysis, and normalized LR weights) were performed.

#### 3.7.1. Normalized Logistic Regression (LR) Weights

Normalized LR weights of the target miRNAs were determined for the four different scenarios discussed earlier (Table 8). In each scenario, the normalized LR weights were normalized such that the summation of the absolute value of the normalized LR weights would be equal to 1. The normalized LR weights indicate the aka contribution of each miRNA to the output of the model. The highest diagnostic potential was shown for miR-150 in the discrimination of HCC patients from healthy controls, non-HCC individuals, and HCV patients with normalized LR weights of 0.21, 0.28, and 0.28, respectively. The negative LR weight of miR-217 indicates a decreasing predicted probability of this miRNA in discriminating HCC patients from non-HCC and HCV patients. However, both miR-150 and miR-424 have the largest normalized weights of the logistic regression model and thus are considered the most important targets that are linearly correlated with the patient’s status. The optimum cut-off probability value was computed for each scenario. The calculated AUC of the miRNAs combined panel proposed to discriminate HCC patients from those with HCV infection was 0.62. The computed sensitivity and specificity were 62.1% and 61.5%, respectively, with accuracy around 62% (Figure 14).

#### 3.7.2. Permutation Feature Importance Measurement

The permutation feature importance measure was used to estimate the most relevant feature to the endpoint by permuting the feature values across different samples with multiple repetitions. The permutation feature importance measure was performed for three scenarios, and the results are presented in box plots (Figure 15). The permutation feature importance measure reported similar results to normalized LR weights, in which miR-150 and miR-424 have the highest predictive probabilities for identifying HCC patients across the three scenarios. Additionally, by comparing HCC and HCV patients, the top five miRNAs with significant importance were miR-150, miR-424, miR-3607, miR-183, and miR-224 in normalized LR weights and permutation feature importance. However, the permutation feature importance measurements reported only three miRNAs (miR-150, miR-424, and miR-199a) capable of discriminating HCC patients from non-HCC individuals. The logistic method failed to produce a boxplot to compare HCV patients and healthy individuals.

#### 3.7.3. SHapley Additive exPlanation (SHAP) Analysis

We constructed the SHAP summary plot to identify the features that influenced the prediction model the most (Figure 16). According to the prediction model, the higher the SHAP value of the miRNA, the more likely it is correlated with the disease. SHAP values for specific features exceeding zero represent an increased risk of HCC development. MiR-217 showed the highest feature values among the four scenarios which were considered. The significant importance of the target miRNAs in relation to HCC disease was slightly different from the other two logistic regression methods. MiR-217, miR-224, miR-199a had the highest SHAP feature values, while miR-150 and miR-424 were ranked 7th and 8th in the correlation with HCC.

### 3.8. Pathway Studio

Assessment of the altered molecular functions and pathways associated with HCC showed that seven targets (miR-142, miR-150, miR-183, miR-199a, miR-217, miR-224, and miR-3607) are involved in cellular pathways related to mRNA degradation and cell migration. Those pathways are known to develop different types of cancers, including liver cancer, neoplasm formation, and metastasis (Figure 17). Proteins involved in the development of HCC are also linked to the pathogenesis of other carcinomas. They are highly associated with breast cancer, renal cell carcinoma, and colorectal carcinoma. Moreover, liver metastasis is directly correlated with the onset of breast cancer (Table 9).

Furthermore, the most relevant targeted pathways of liver carcinogenesis are hepatitis C infection, HCC, and metastasis (Figure 18). Our pathway analysis showed that miR-183, miR-199a, miR-215, and miR-224 play a significant regulatory role in HCC development, while the sequence of miR-3607 exhibits a binding site that inhibits the progression of HCC. Alteration in miR-199a expression affects cellular functions and indirectly contributes to liver carcinogenesis by regulating HCV infection. While the expression of miR-183 is dysregulated in liver cirrhosis, premalignant lesions, and HCC, it directly inhibits the tumor suppressor gene in liver carcinogenesis. MiR-224 exhibits overexpression in HCC and contributes to the development of liver cancer.

## 4. Discussion

HCC is considered one of the widely spread and most aggressive malignancies worldwide, representing more than 80% of primary liver cancers [37]. HCV virus plays a major role in hepatic carcinogenicity [38], accounting for 25% of the HCC cases worldwide [39]. Generally, the poor prognosis of HCC and low survival rates drive the global efforts to identify novel biomarkers for early detection of the disease [40]. The weak diagnostic potential of AFP in detecting HCC patients was highlighted in our study, in which 61% of the enrolled HCC patients had normal AFP levels (<20 ng/mL). One reason could be the relatively low sensitivity of AFP in detecting HCC patients at the early stages of the disease (BCLC stage 0, A, and B is 18%, 64%, and 18% of the enrolled patients, respectively), in addition to the FL size (<3 cm in 40.7% of the patients) and number (single FL in 64% of the HCC patients). This finding was reported in previous studies [41,42]. Thus, this research study aims to identify a miRNAs panel to serve as a non-invasive biomarker for predicting HCC in chronic HCV patients.

The results obtained from our bioinformatics analysis highlighted four overlapping miRNAs sharing the same expression patterns between the GEO microarray and TCGA datasets. MiR-224 was upregulated, while miR-150, miR-199b, and miR-424 were downregulated in liver tissues. In addition to these four non-coding RNAs, analysis of TGCA datasets resulted in five DE-miRNAs, two upregulated miRNAs (miR-183 and miR-217), and three downregulated miRNAs (miR-142, miR-215, and miR-3607) in HCC liver tissues. The nine candidate miRNAs obtained from the TCGA dataset analysis were chosen to validate their serum expression further using real-time PCR (qPCR).

Real-time PCR results in this study showed a significant increase in the serum concentration of the nine-candidate miRNAs in HCC patients and HCV patients relative to healthy individuals (*p* < 0.0001). However, the expression levels of miR-424, miR-199a, miR-142, and miR-224 were significantly altered in HCC patients upon comparison with the non-cirrhotic subjects. At the same time, miR-199a and miR-183 showed differential expression in HCV cirrhotic patients relative to non-cirrhotic subjects. The only target that showed a significant difference between HCV patients with LC and HCC patients was miR-424, predicting diagnostic and prognostic biomarker powers for these miRNAs in HCC.

Serum expression of miR-183, miR-215, and miR-224 was previously studied in HCC, and the results were consistent with our findings. Several research studies reported a significant increase in the miR-183 concentration in the serum of cirrhotic and HCC patients relative to healthy individuals. They also concluded that the sensitivity and specificity of using miR-183 as a biomarker for HCC detection were 57.9% and 76.2% in the serum, respectively, proposing a diagnostic potential of miR-183 in differentiating HCC patients from those with LC without malignancy [43,44]. Additionally, the serum miR-183 level in HCC patients after surgery was significantly lower compared to the expression before surgery [45], confirming that the increase in miR-183 serum levels is positively correlated with the presence of HCC FLs. Furthermore, elevated serum expression levels of miR-224 were explained in several studies [46,47,48]. The serum miR-224 level was correlated with AFP levels and other serum parameters, indicating liver damage and an association with poor survival. The miR-224 increased serum expression was also correlated with the BCLC stage progression. Higher miR-224 expression was recorded in patients with BCLC stage C than stage B. Therefore, miR-224 concentration could be BCLC stage-dependent, in addition to being a prognostic biomarker for HCC patients’ survival [49,50]. In a previously published study conducted on the Egyptian population, overexpression of miR-224 and miR-215 in the serum of HCV-HCC Egyptian patients compared to healthy individuals was detected using qPCR [51,52]. Although the increase in miR-215 serum levels failed to distinguish between HCV, HBV, and HCC patients, its expression was significantly increased in all groups relative to healthy controls [53]. However, multiple recent studies relied on the serum miR-215 expression levels to differentiate between patients with fibrosis and those with LC and between HCC patients and other hepatic disease patients. It was also observed that miR-215 expression was positively correlated with HCV viral load [54,55], suggesting that miR-215 might act as a potential prognostic biomarker for liver disease.

Furthermore, contradicting results were reported about the serum expression of miR-150, miR-199a, and miR-424 in HCC. In a previously published study, the increase in miR-150 serum levels in HCV-HCC patients in African Americans and Caucasians relative to healthy controls was comparable to our findings. Moreover, a significant increase in the serum levels was observed in the HCV cirrhotic group upon comparison to healthy individuals in both ethnic groups [56]. However, analysis of miR-150 serum expression in HCV-HCC Egyptian patients in a different study recorded a significant decrease in serum miR-150 levels in HCC patients compared to healthy controls and non-cirrhotic HCV patients, and no significant difference was found between HCC and cirrhotic HCV patients. Moreover, the expression levels were lower in HCV cirrhotic patients compared to non-cirrhotic individuals. These results contradict our findings, as miR-150 serum expression was significantly higher in the HCC group than controls, non-cirrhotic, and cirrhotic subjects, whereas no significant difference was observed between non-cirrhotic and cirrhotic patients nor between cirrhotic and HCC individuals [57]. Although few published studies addressed the regulation of miR-424 serum expression in HCC, their results were inconsistent. A significant increase in serum miR-424-3p levels in HCC patients compared to healthy control was reported in a previous study [58].

On the other hand, analysis of the miR-424 serum expression using qPCR showed that its expression was reduced in HCC patients compared to healthy individuals in another. The decreased expression was also correlated with serum AFP levels, vein invasion, and progression of the tumor, nodes, and metastasis [59]. Interestingly, qPCR results in a third study failed to find a significant difference in serum miR-424 levels in HCC patients relative to the healthy controls [60]. As for miR-199a, multiple research studies on Egyptian patients concluded that serum miR-199a overexpression in severe chronic hepatic inflammation and HCV genotype-4 patients occurs, especially in late-stage fibrosis compared to early fibrotic stages. This may be due to the induced inflammation triggered by HCV to the hepatocytes, concluding that members of the miR-199 family are linked to liver fibrosis progression in HCV patients [52,61,62].

On the other hand, other studies reported a reduction in miR-199a expression in the serum of HCC patients [61,62,63,64,65]. It was observed that the decrease in miR-199a serum expression was inversely proportional to apoptotic markers such as programmed cell death protein four and cytochrome C [62].

Additionally, the antiviral activity of miR-199a against HCV was proved. The inhibition mechanism is attributed to the interaction between miR-199a and the step loop II region at the 5′UTR of HCV, resulting in inhibition of HCV replication. Thus, based on their findings, increased expression of miR-199a was associated with cell cycle arrest, suppression of cellular invasion improving sensitivity to chemotherapy, and hinderance of HCV genome replication [66,67,68]. Therefore, we suggest further analysis of the differential expression of miR-150, miR-424, and miR-199a serum levels in HCV and HCC patients.

To the best of our knowledge, this research is the first to report circulatory differential expression of miR-142, miR-217, and miR-3607 in the serum of HCV and HCC patients. In a previous study, miR-142-5p serum levels were inversely correlated with the serum albumin levels in HCV-HCC patients, although the differential expression was not reported [69]. Downregulation of miR-3607 was reported in HCC tissues, and it was suggested that miR-3607 interferes with the epithelial to mesenchymal transition resulting in inhibition of tumor proliferation, migration, and invasion. It might be considered a potential biomarker for HCC progression [70,71,72]. It is worth mentioning that 20% of our qPCR signals in serum miR-217 amplification were undetectable. Thus, these samples were normalized by replacing their Ct values by the maximum allowed number of cycles = 40 [31] to avoid losing a significant number of samples that showed true amplification with other targets in the data analysis and have a consistent samples’ number for all miRNAs. Therefore, based on the scarce data on circulatory miR-217 expression in the literature and our findings, we suggest that miR-217 serum differential expression might not affect the regulation of HCC progression. The highly significant positive correlations (*p* < 0.0001) that correlate with HCV- cirrhotic and HCC groups suggest the direct involvement of the target miRNAs in HCC disease development/progression.

The polygenic nature of HCC and the complexity of serum as a detection platform favored the use of a multiple biomarkers approach over a single one [73]. We proposed a novel miRNAs study panel that could play a pivotal role in HCC detection in the current study. A combined panel of 5 miRNAs dramatically increased the sensitivity and specificity of distinguishing HCC patients from healthy individuals to 100% and 95.12%, respectively, with 97.85% detection accuracy. A similar trend was obtained in identifying HCV patients to reach 90.5% sensitivity, 85.37% specificity, and 89% accuracy upon relying on a 5-miRNAs-combined panel. The success chance of using multiple miRNAs in a single panel was manifested in discriminating HCC patients from non-HCC individuals in both the SVR and the treatment naïve groups. A combined five miRNAs panel enhanced the sensitivity and specificity of detection to 83.3% and 63.73%, respectively, in the SVR groups. At the same time, the 6-miRNAs combined panel provided better results in the treatment naïve patients (89.3% sensitivity and 72.6% specificity).

Moreover, the combined panel was successfully used to assess the accuracy of distinguishing HCC from HCV patients. Only three (miRNAs 424, miR-199a, and miR-150) showed statistically significant ROC AUC values. A combined panel of 2 miRNAs did not improve the detection sensitivity and specificity (61.54% and 56.84%) compared to AFP (62.3% and 64.57%). However, in comparing HCV and HCC patients belonging to the SVR groups, the AUC values of miR-424, miR-142, and miR-3607 were statistically significant. Interestingly, the 2-miRNAs combined panel ameliorated the sensitivity and specificity in the SVR group to 70.83% and 61.9%, respectively, in comparison to AFP results (51.43% and 60.67%, respectively). Nevertheless, the inclusion of AFP to the miRNAs combined panel improved the sensitivity and specificity of detection in the HCC-SVR patients to 70.83% and 73.02%, respectively. However, no remarkable difference was observed in the HCC unclassified group.

Three logistic regression analyses were performed to validate the predictive ability of the individual miRNAs and the combined panels obtained by the ROC analysis. The results obtained from normalized LR weights and permutation feature importance measures were closely related. MiR-150, miR-424, miR-3607, miR-183, and miR-224 were the top five miRNAs with the highest predictability in discriminating HCC patients from HCV-infected patients in both methods, with sensitivity and specificity of 62% and 61.5%, respectively. These findings were comparable to the results obtained from the ROC analysis performed on the qPCR results, in which miR-150, miR-424, and miR-199a had significant AUC, and the combined panel had 61.5% sensitivity and 57% specificity in the identification of HCC patients from HCV infected patients. However, the AUC of miR-424, miR-3607, and miR-142 had significant *p* values, with a sensitivity of 70.8% and specificity of 62% for the combined panel upon comparing HCC-SVR patients with HCV-SVR ones. On the other hand, the results obtained from the SHAP analysis were different from the other methods. Consequently, we used the intersection of all these analyses to represent the most important miRNAs composing the combined panel.

Cell migration and mRNA degradation were two untargeted pathways regulated by seven DE-miRNAs under study (miR-142, miR-150, miR-183, miR-199a, miR-217, miR-224, and miR-3607), implicating the development of liver cancer. Moreover, our findings revealed that the dysregulation in the differential expression of six miRNAs (miR-150, miR-183, miR-199a, miR-215, miR-224, and miR-3607) was primarily involved in the pathogenesis of HCC. Different mechanisms might explain the alteration in the affected pathways, such as the upregulation of miR-224, which was recorded in HCC through activating the tumor necrosis factor-alpha, lipopolysaccharide, and lymphotoxin-alpha inflammatory pathways, resulting in cell migration/invasion in the HCC [74], while miR-199a was downregulated in HCC, inducing G1-phase cell cycle arrest and decreasing the invasiveness via targeting c-Met and mTOR [75]. MiR-183 was reported to inhibit apoptosis in human HCC cells by repressing the pro-apoptotic molecule PDCD4 expression [76]. Identifying the altered cellular pathways in HCC will facilitate the prediction of the corresponding gene targets, which might act as potential biomarkers in HCC prognosis and establish targets for intervention.

Finally, we believe that the choice of the candidate miRNAs within the panel provided a multifunctional tool for HCC early detection. MiR-199a could act as a marker for liver fibrosis progression, while a prediction for HCC poor prognosis could be achieved via miR-224. While miR-424 might provide a prediction tool for tumor recurrence, miR-3607 is proposed to play a significant role in the prognosis of HCC via inhibition of tumor invasiveness, proliferation, and recurrence, suggesting a potential prognostic biomarker ability for these miRNAs. Moreover, the inclusion of miR-150 might provide information about the fibrosis and cirrhosis progression in HCV patients. Thus, using the miRNAs combined panel will facilitate and improve HCC prognosis more than a conventional single biomarker approach.

### Study Limitations

The study limitations include a relatively low sample size of the enrolled subjects in the study and the heterogeneous cohort expressed in different HCV treatment options in HCV-SVR and HCC-SVR groups. The choice of the endogenous reference gene (SNORD 68) in qPCR amplification and data analysis was made following the previous research recommendations in the literature. However, data normalization using an endogenous reference panel could have been conducted. Furthermore, the variation in Ct values among the technical replicates could be attributed to the low cDNA template concentration used in the qPCR amplification reaction. However, the used concentration was approximately close to the upper recommended range by the kit’s manufacturer. The lack of AFP measurements in the healthy control samples hindered comparing the efficacy of the miRNAs panel versus AFP in HCC and HCV detection.

## 5. Conclusions

Nearly 80% of HCC cases are untreatable due to the patients’ presentation at their advanced stages. Consequently, the determination of a non-invasive biomarker would hasten and promote the diagnosis of HCC, decrease the risks of surgical intervention, and permit the non-invasive monitoring resulting in better opportunities for therapeutic options. In this study, the serum differential expression of nine miRNAs (miR-142, miR-150, miR-183, miR-199a, miR-215, miR-217, miR-224, miR-424, and miR-3607) was significantly overexpressed in HCC and HCV patients compared to healthy individuals. Using a combined panel of five miRNAs (miR-150, miR-199a, miR-224, miR-424, and miR-3607) improved the overall sensitivity and specificity of HCC detection. It could serve as a promising candidate for an early prediction tool for HCC in chronic HCV patients. However, it is essential to conduct a large-scale population-based study to evaluate the prognostic value of the miRNAs panel.

## Figures and Tables

**Figure 1 cancers-14-03036-f001:**
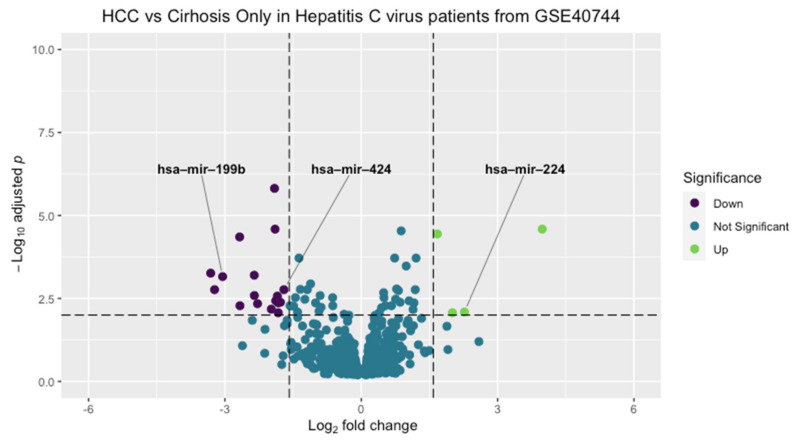
Volcano plot of microarray differentially expressed miRNAs. MiRNAs colored in purple were downregulated, and those colored in green were upregulated in HCC samples compared to cirrhotic HCV samples. The filtering criterion was adjusted using a log_2_ fold change cutoff of 1.5 and an adjusted *p*-value cutoff of 0.01. (GEO: Gene expression omnibus, HCC: Hepatocellular carcinoma).

**Figure 2 cancers-14-03036-f002:**
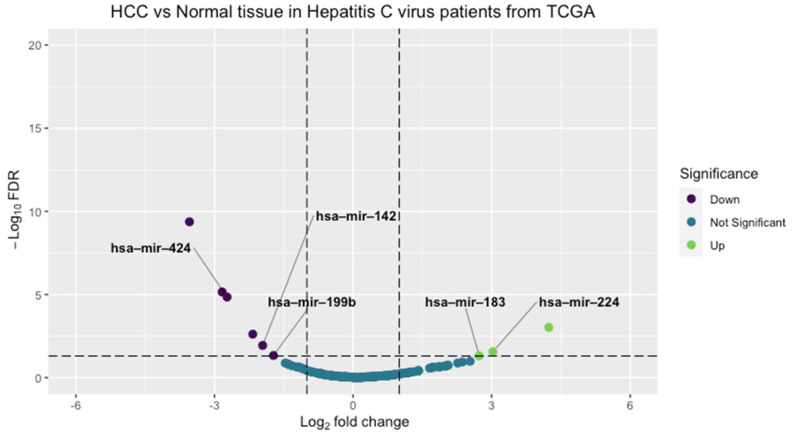
Volcano plot of TCGA differentially expressed miRNAs. MiRNAs colored in purple were downregulated, and those colored in green were upregulated in HCC samples compared to non-tumor tissue samples. The filtering criterion was adjusted using a log_2_ fold change cutoff of 1 and FDR cutoff of 0.05. (FDR: False discovery rate, HCC: Hepatocellular carcinoma, TCGA: The cancer genome atlas).

**Figure 3 cancers-14-03036-f003:**
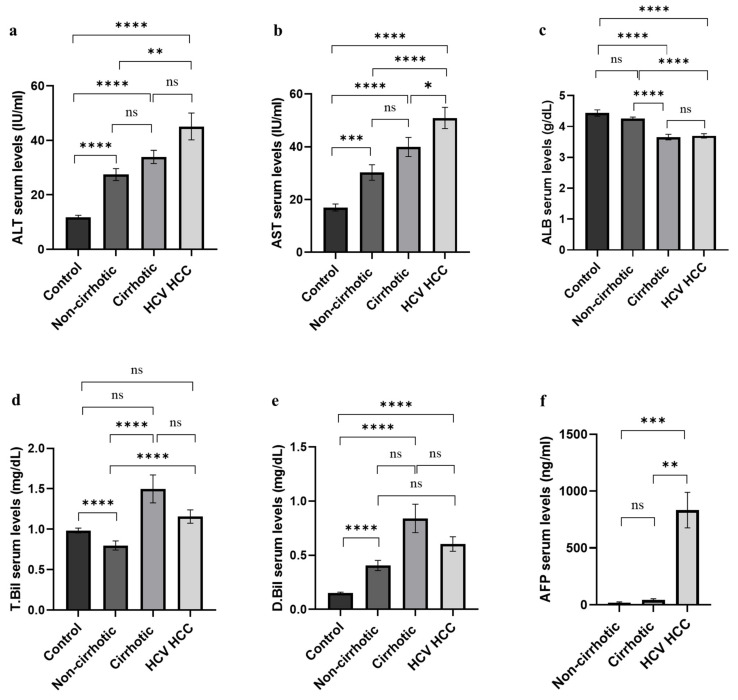
Mean of serum concentration of liver biomarkers in the study groups. (**a**) Comparison of mean values of (**a**) ALT, (**b**) AST, (**c**) albumin, (**d**) total bilirubin, (**e**) direct bilirubin, and (**f**) AFP. Values are expressed as mean ± SEM. Statistical significance (**** indicates *p* ≤ 0.0001, *** indicates *p* ≤ 0.001, ** indicates *p* ≤ 0.01, * indicates *p* ≤ 0.05, “ns” indicates statistically non-significance). (AFP: alpha-fetoprotein, ALB: albumin, ALT: alanine aminotransferase, AST: aspartate aminotransferase, D. Bil: direct bilirubin, T. Bil: total bilirubin).

**Figure 4 cancers-14-03036-f004:**
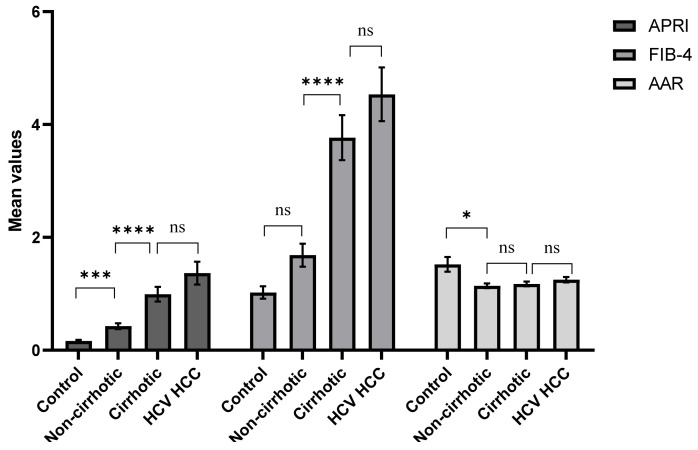
Mean values of the non-invasive indices APRI, FIB-4, and AAR in the study groups. Comparison of the statistical significance of the three non-invasive indices in differentiation between different disease stages. Values are expressed as mean ± SEM. Statistical significance (**** indicates *p* ≤ 0.0001, *** indicates *p* ≤ 0.001, * indicates *p* ≤ 0.05, ns indicates non-significance). (AAR: AST to ALT ratio, APRI: Aspartate aminotransferase to platelet ratio index, FIB-4: fibrosis-4 index).

**Figure 5 cancers-14-03036-f005:**
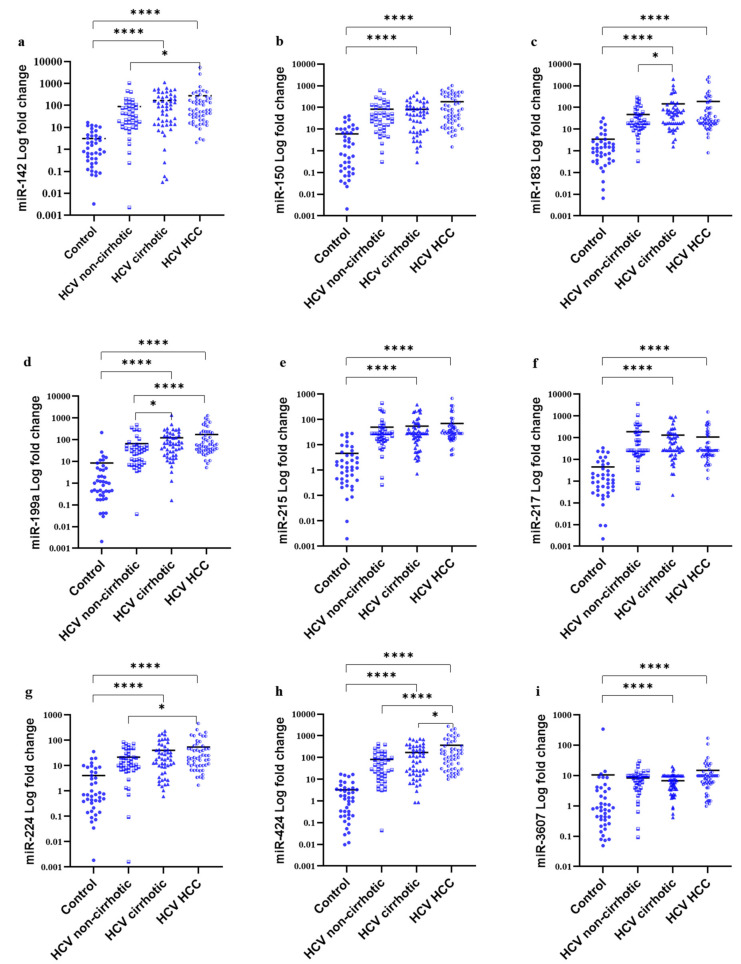
Fold change of the differentially expressed miRNAs in the study groups. Scatter dot plots demonstrating the fold change of serum expression of the target miRNAs [(**a**) miR-142 (**b**) miR-150, (**c**) miR-183 (**d**) miR-199a (**e**) miR-215 (**f**) miR-217 (**g**) miR-224 (**h**) miR-424 and (**i**) miR-3607] among the study groups. Y-axis represents the log of the fold change of each miRNA; X-axis shows the study groups. * Corresponds to significant *p-*value ≤ 0.05, **** corresponds to significant *p-*value < 0.0001. Each experiment was performed in duplicates (HCC: Hepatocellular carcinoma).

**Figure 6 cancers-14-03036-f006:**
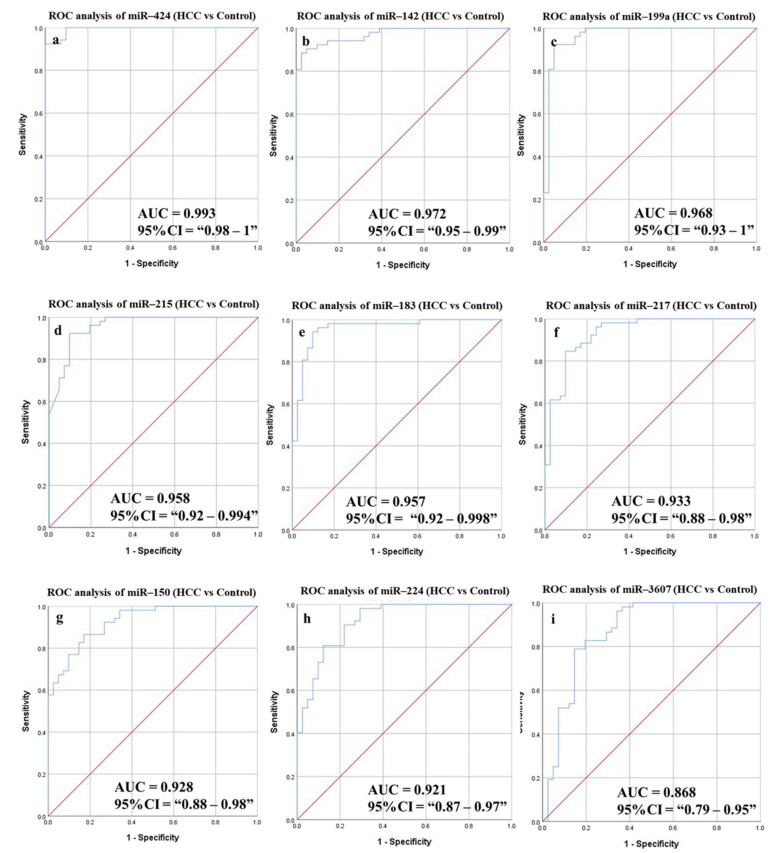
ROC curves and AUC for the DE-miRNAs differentiate between HCC patients and healthy individuals. The diagnostic potential and AUC of nine DE-miRNAs [(**a**) miR-424 (**b**) miR-142 (**c**) miR-199a (**d**) miR-215 (**e**) miR-183 (**f**) miR217 (**g**) miR150 (**h**) miR-224 and (**i**) miR-3607] were calculated. (DE: Differentially expressed, HCC: Hepatocellular carcinoma, ROC: Receiver operating characteristic).

**Figure 7 cancers-14-03036-f007:**
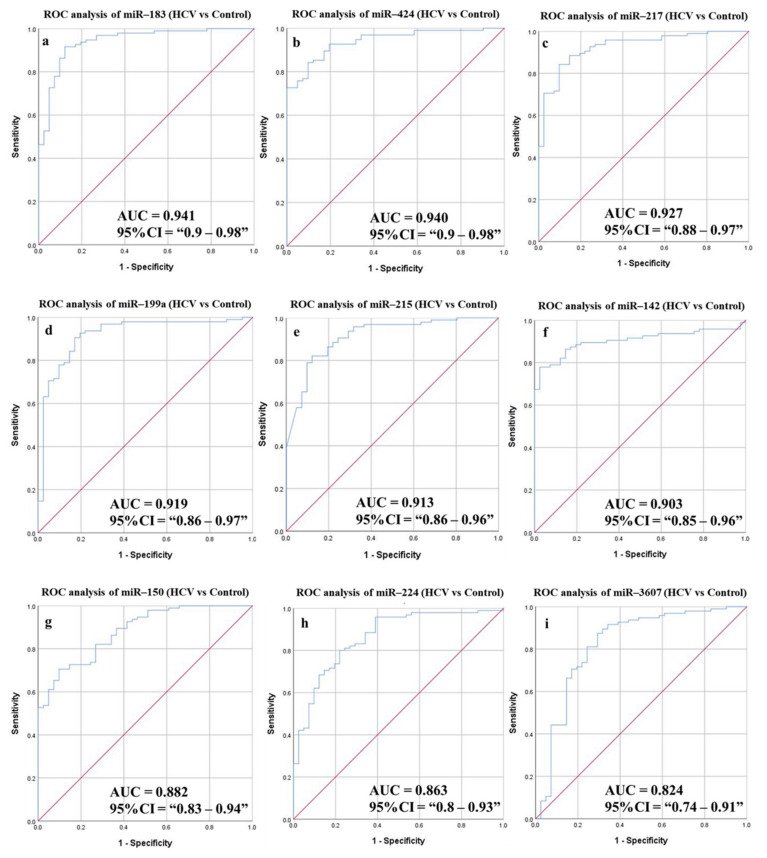
ROC curves and AUC for the DE-miRNAs to differentiate between HCV patients and healthy individuals. The diagnostic potential and AUC of nine DE-miRNAs [(**a**) miR-183 (**b**) miR-424 (**c**) miR-217 (**d**) miR-199a (**e**) miR-215 (**f**) miR-142 (**g**) miR-150 (**h**) miR-224 and (**i**) miR-3607] were calculated. (DE: Differentially expressed, HCC: Hepatocellular carcinoma, ROC: Receiver operating characteristic).

**Figure 8 cancers-14-03036-f008:**
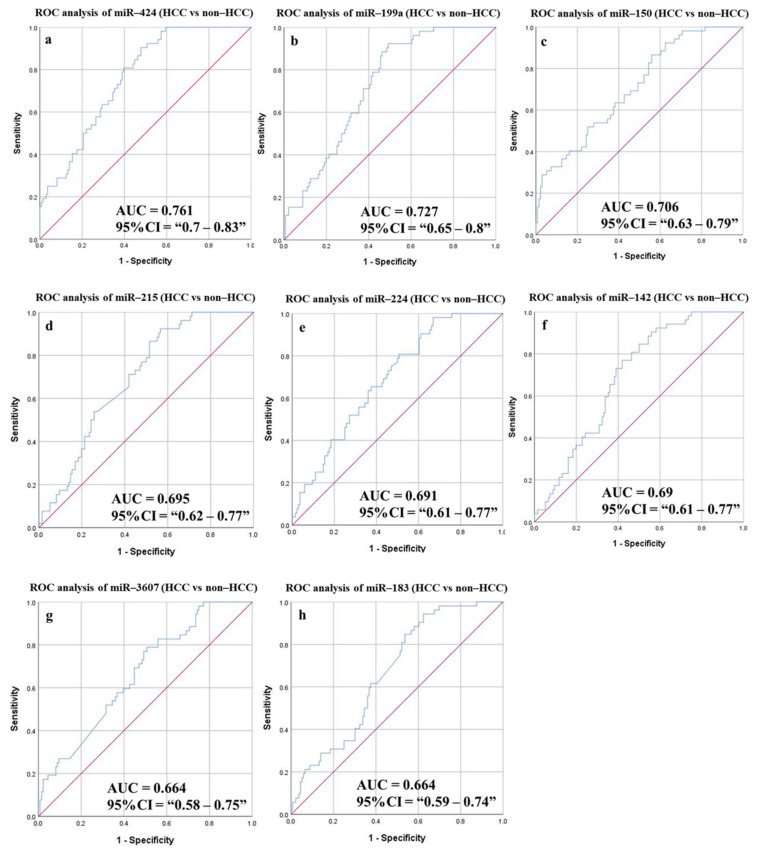
ROC curves and AUC for DE-miRNAs differentiate between HCC patients and non-HCC (healthy controls, non-cirrhotic, and cirrhotic HCV patients). The diagnostic potential and AUC of the eight DE-miRNAs [(**a**) miR-424 (**b**) miR-199a (**c**) miR-150 (**d**) miR-215 (**e**) miR-224 (**f**) miR-142 (**g**) miR-3607 and (**h**) miR-183] were calculated. (DE: Differentially expressed, HCC: Hepatocellular carcinoma, ROC: Receiver operating characteristic).

**Figure 9 cancers-14-03036-f009:**
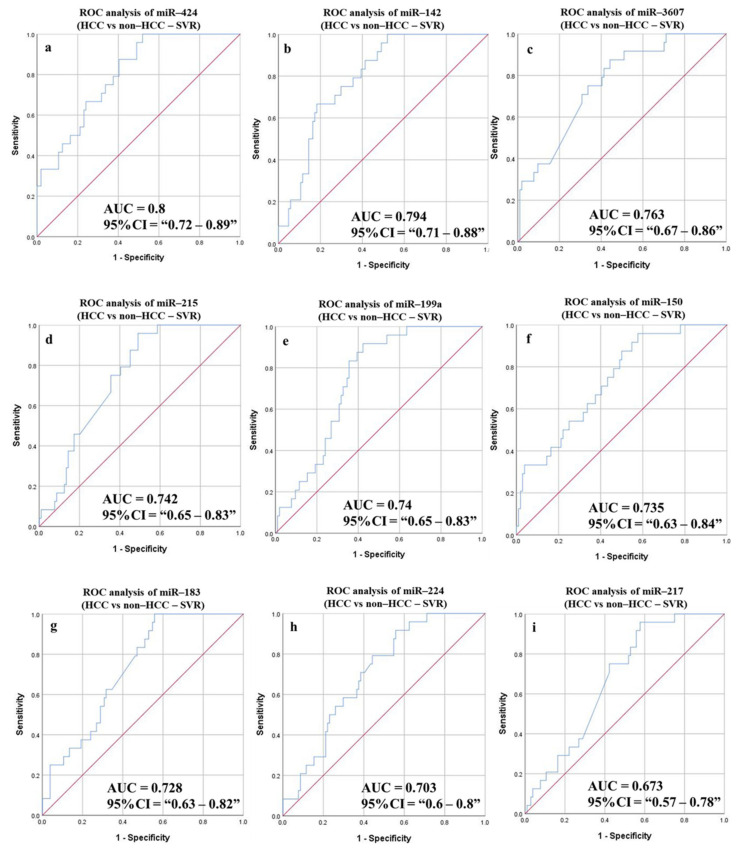
ROC curves and AUC for the DE-miRNAs to differentiate between HCC (SVR) patients and non-HCC (healthy controls, non-cirrhotic (SVR), and cirrhotic (SVR) HCV patients). The diagnostic potential and AUC of nine DE-miRNAs [(**a**) miR-424 (**b**) miR-142 (**c**) miR-3607 (**d**) miR-215 (**e**) miR-199a (**f**) miR-150 (**g**) miR-183 (**h**) miR-224 and (**i**) miR-217] were calculated.

**Figure 10 cancers-14-03036-f010:**
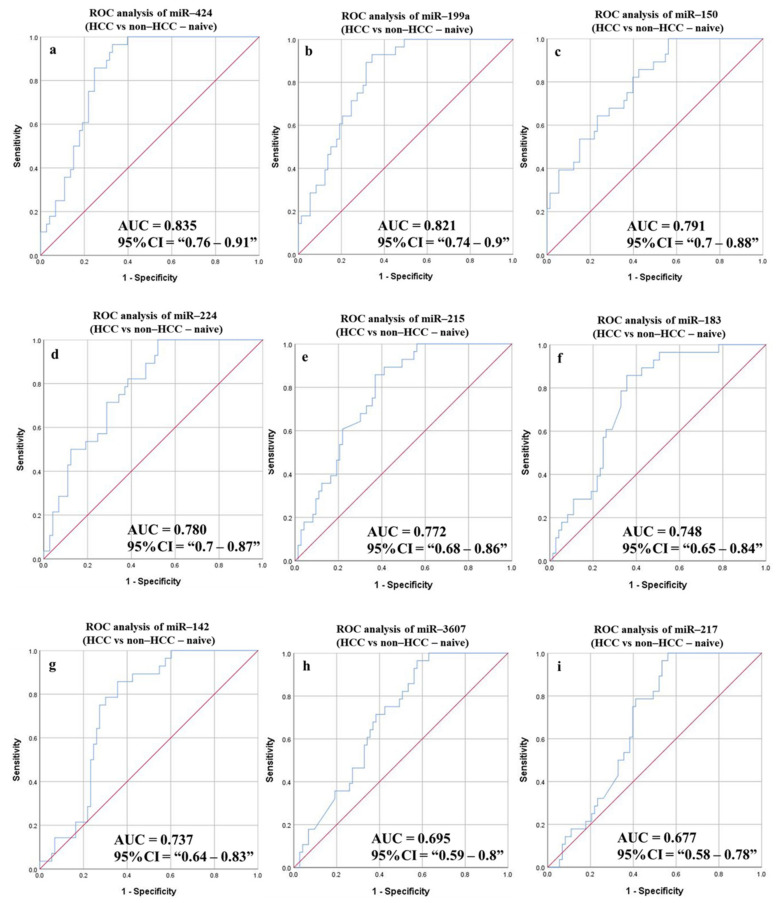
ROC curves and AUC for the DE-miRNAs to differentiate between HCC (treatment naive) patients and non-HCC (healthy controls, non-cirrhotic (treatment naive), and cirrhotic (treatment naive) HCV patients). The diagnostic potential and AUC of nine DE-miRNAs [(**a**) miR-424 (**b**) miR-199a (**c**) miR-150 (**d**) miR-224 (**e**) miR-215 (**f**) miR-183 (**g**) miR-142 (**h**) miR-3607 and (**i**) miR-217] were calculated. (DE: Differentially expressed, HCC: Hepatocellular carcinoma, ROC: Receiver operating characteristic).

**Figure 11 cancers-14-03036-f011:**
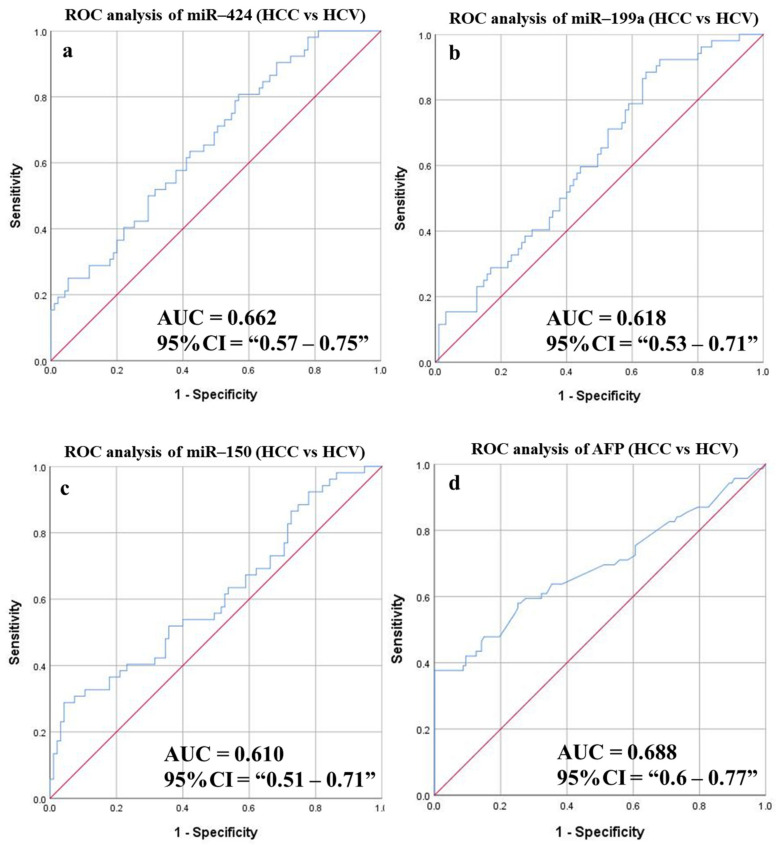
ROC curves and AUC for DE-miRNAs compared to AFP to differentiate between HCC and HCV patients. The diagnostic potential and AUC of three DE-miRNAs [(**a**) miR-424 (**b**) miR-199a and (**c**) miR-150], in addition to (**d**) AFP were calculated. (AFP: Alpha-fetoprotein, DE: Differentially expressed, HCC: Hepatocellular carcinoma, ROC: Receiver operating characteristic).

**Figure 12 cancers-14-03036-f012:**
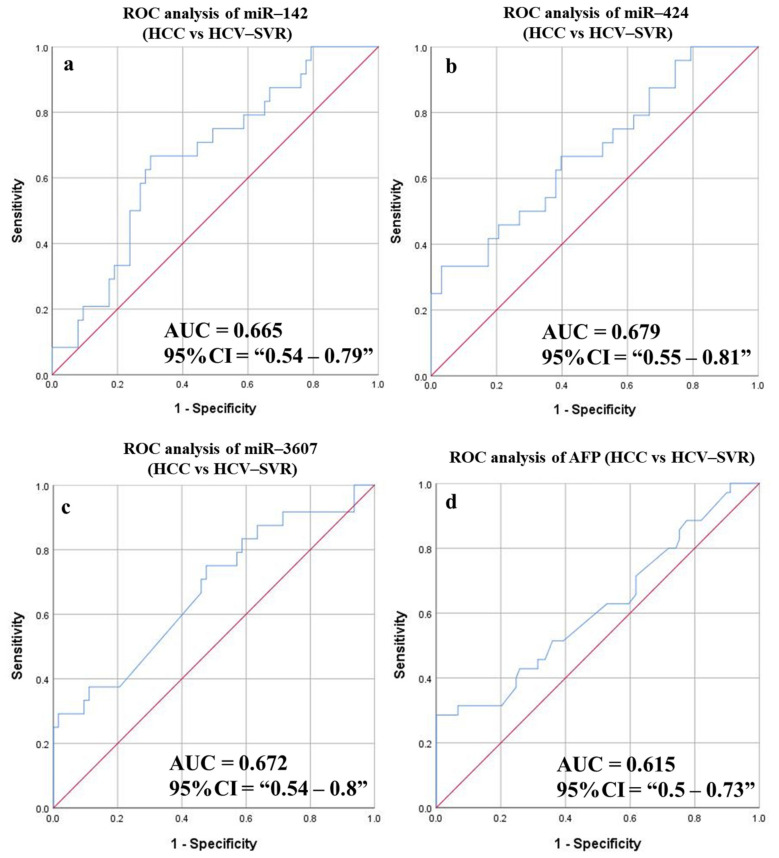
ROC curves and AUC for DE-miRNAs compared to AFP to differentiate between HCC (SVR) and HCV (SVR) patients. The diagnostic potential and AUC of three DE-miRNAs [(**a**) miR-142 (**b**) miR-424 and (**c**) miR-3607], in addition to (**d**) AFP, were calculated. (AFP: Alpha-fetoprotein, DE: Differentially expressed, HCC: Hepatocellular carcinoma, ROC: Receiver operating characteristic).

**Figure 13 cancers-14-03036-f013:**
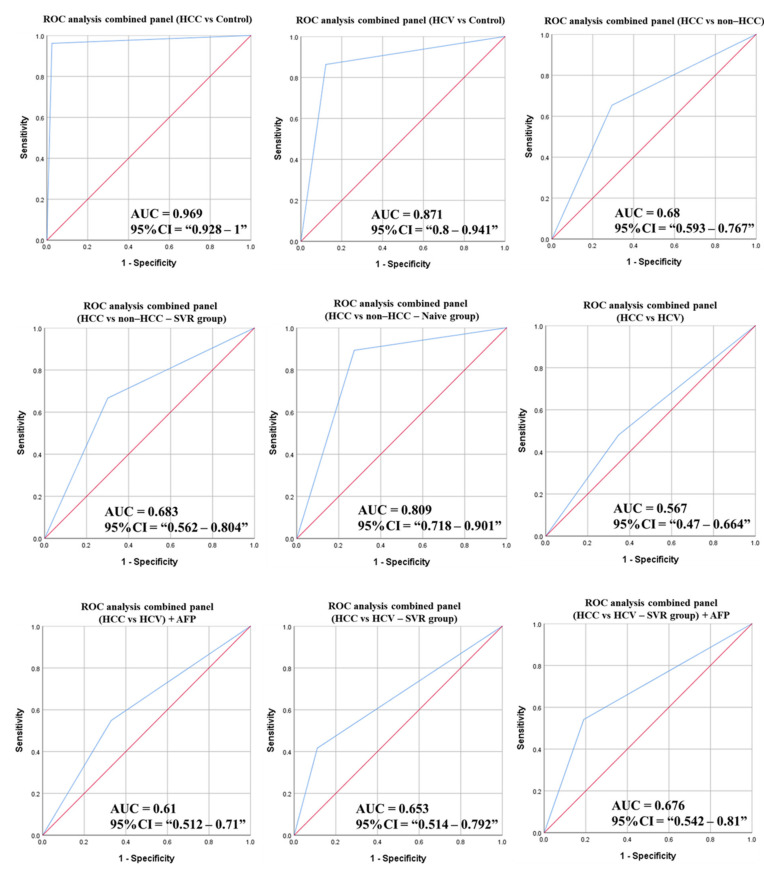
ROC curves and AUC for the combined panels of the DE-miRNAs among the study groups. (AFP: Alpha-fetoprotein, DE: Differentially expressed, HCC: Hepatocellular carcinoma, ROC: Receiver operating characteristic, SVR: Sustained virological response).

**Figure 14 cancers-14-03036-f014:**
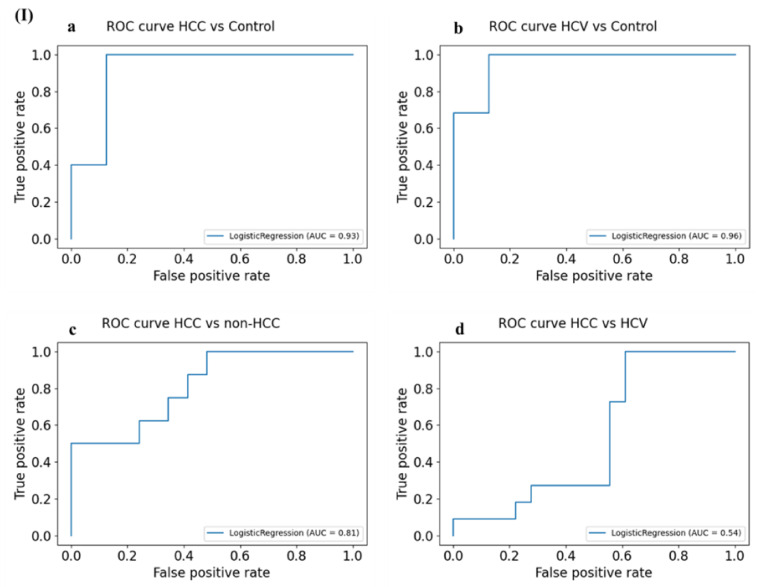
ROC analysis for the predictive diagnostic ability of the target miRNAs combined panels using logistic regression analysis. (**I**) ROC curves of the miRNAs combined panel comparing (**a**) HCC patients to healthy individuals, (**b**) HCV patients to healthy individuals, (**c**) HCC patients to non-HCC individuals, and (**d**) HCC to HCV infected patients. (**II**) Plot of the computed sensitivity, specificity, and accuracy highlighting the predictive ability of the miRNAs combined panels in (**a**) HCC patients to healthy individuals, (**b**) HCV patients to healthy individuals, (**c**) HCC patients to non-HCC individuals, and (**d**) HCC to HCV infected patients.

**Figure 15 cancers-14-03036-f015:**
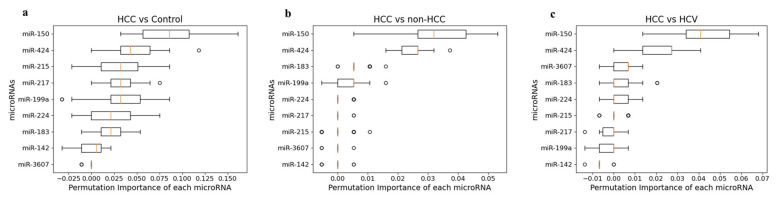
Boxplots showing the permutation importance of each miRNA. Feature importance analyses were performed to assess the predictive probability of each miRNA in discriminating (**a**) HCC patients from controls, (**b**) HCC patients from non-HCC individuals, (**c**) HCC from HCV infected patients.

**Figure 16 cancers-14-03036-f016:**
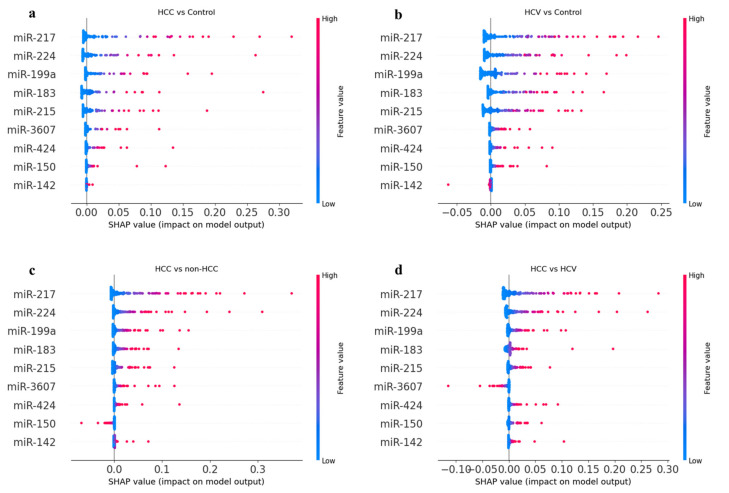
SHAP summary plot describing the impact of target miRNAs on the model output. The higher the SHAP value of a feature, the higher the probability of disease correlation. SHAP values were used to provide consistent and locally accurate attribution values for each miRNA within each prediction model; (**a**) HCC patients compared to healthy individuals, (**b**) HCV infected patients compared to healthy individuals, (**c**) HCC patients compared to non-HCC individuals, and (**d**) HCC compared to HCV infected patients.

**Figure 17 cancers-14-03036-f017:**
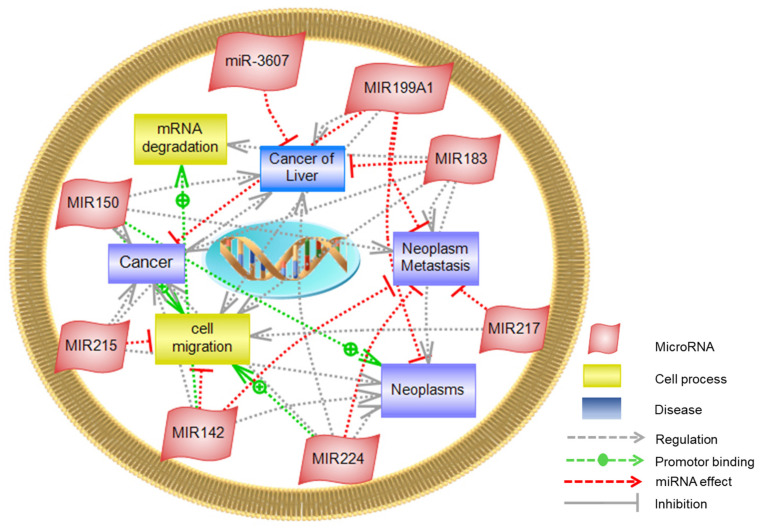
Altered miRNA-Molecular functions and Biological Processes. The global untargeted affected pathways involving: cell migration and mRNA degradation resulting in different cancer types, especially liver cancer, neoplasms, and neoplasm metastasis, implicated in the miRNAs’ alteration (miR-142, miR-150, miR-183, miR-199a, miR-217, miR-224, and miR-3607).

**Figure 18 cancers-14-03036-f018:**
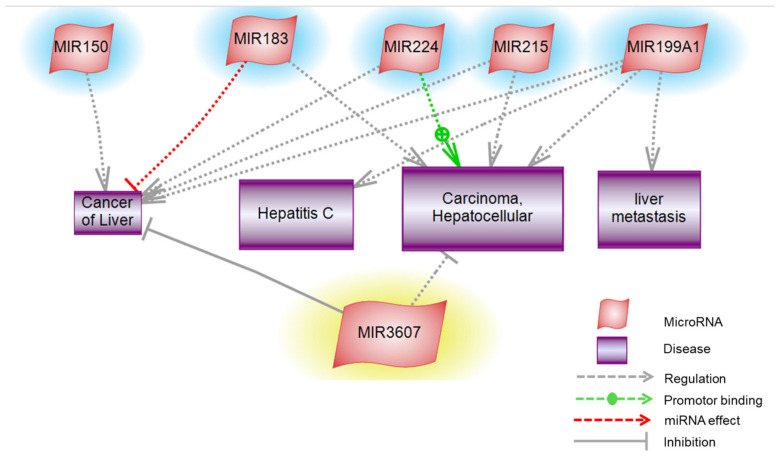
Targeted analysis of Molecular functions and Biological Processes. The targeted affected pathways involving: Hepatitis C, HCC, other types of liver cancer, and liver metastasis implicated in the miRNAs’ alteration (miR-150, miR-183, miR-199a, miR-215, miR-224, and miR-3607).

**Table 1 cancers-14-03036-t001:** Clinicopathological and demographic features of the study population.

Clinico- Pathological Features	No. of Participants (*n* = 245)	Groups					Statistics	
		Control (*n* = 44) (% within Group)	HCV Non-Cirrhotic (*n* = 62) (% within Group)	HCV Cirrhotic (*n* = 67) (% within Group)	HCV-HCC (*n* = 72) (% within Group)	Test Statistics	*p-*Value (among the Groups)	*p-*Value (HCV vs. HCC)
Age	-	54.6 ± 1.93	52.6 ± 1.253	59.2 ± 1.45	61.4 ± 0.96	26.53 ^a^	<0.0001	0.388 ^d (N.S)^
Mean age (≤57)	136	25 (56.8%)	48 (77.4%)	36 (53.7%)	27 (37.5)	21.624 ^b^	<0.0001	-
Mean age (>57)	109	19 (43.2%)	14 (22.6%)	31 (46.3%)	45 (62.5%)	*-*	-	-
Gender	-	-	-	-	-	43.775 ^b^	<0.0001	<0.0001
Male	122	30 (68.2%)	15 (24.2%)	24 (35.8%)	53 (73.6%)	*-*	-	-
Female	123	14 (31.8%)	47 (75.8%)	43 (64.2%)	19 (26.4%)	*-*	-	-
HCV infection	-	-	-	-		245 ^b^	NA	0.005
Negative	44	44 (100%)	0	0	0	*-*	-	-
Positive (SVR)	126	0	48 (77.4%)	42 (62.7%)	36 (50%)	*-*	-	-
Positive (Treatment naïve)	75	0	14 (22.6%)	25 (37.3%)	36 (50%)	-	-	-
Cirrhosis	-	-	-	-	-	245 ^b^	NA	NA
Negative	106	44 (100%)	62 (100%)	0	0	*-*	-	-
Positive	139	0	0	67 (100%)	72 (100%)	*--*	-	-
ALT		12 ± 0.7	27 ± 2.2	34 ± 2.4	45 ± 5	90.3 ^a^	<0.0001	0.179 ^d (N.S)^
≤40 IU/L	187	42 (100%)	51 (83.6%)	50 (77%)	44 (63%)	22.75 ^b^	NA	-
>40 IU/L	51	0	10 (16.4%)	15 (23%)	26 (37%)		-	-
Missing	7	2	1	2	2	-	-	-
AST	-	17 ± 1.3	30 ± 3	40 ± 3.6	51 ± 4	70.1 ^a^	<0.0001	0.046
≤40 IU/L	167	41 (97.6%)	51 (83.6%)	39 (60%)	36 (51.4%)	35.34 ^b^	<0.0001	-
>40 IU/L	71	1 (2.4%)	10 (16.4%)	26 (40%)	34 (48.6%)	*-*	-	-
Missing	7	2	1	2	2	-	-	-
ALB		4.4 ± 0.1	4.3 ± 0.5	3.7 ± 0.1	3.7 ± 0.1	23.82 ^c^	<0.0001	0.981 ^e (N.S)^
>4 g/dL	122	28 (66.7%)	51 (83.6%)	22 (33.8%)	21 (30.4%)	49.41 ^b^	<0.0001	-
≤4 g/dL	115	14 (33.3%)	10 (16.4%)	43 (66.2%)	48 (69.6%)	*-*	-	-
Missing	8	2	1	2	3	-	-	-
T. Bil		1 ± 0.03	0.8 ± 0.06	1.2 ± 0.2	1 ± 0.1	33.12 ^a^	<0.0001	0.774 ^d (N.S)^
≤1.25 mg/dL	191	41 (97.6%)	57 (93.4%)	44 (67.7%)	49 (71%)	25.19 ^b^	<0.0001	-
>1.25 mg/dL	46	1 (2.4%)	4 (6.6%)	21 (32.3%)	20 (29%)	*-*	-	-
Missing	8	2	1	2	3	-	-	-
D. Bil		0.1 ± 0.01	0.4 ± 0.05	0.8 ± 0.13	0.6 ± 0.1	71.23 ^a^	<0.0001	0.749 ^d (N.S)^
≤0.35 mg/dL	134	42 (100%)	33 (54.1%)	27 (41.5%)	32 (46.4)	41.29 ^b^	NA	-
>0.35 mg/dL	103	0	28 (45.9%)	38 (58.5%)	37 (53.6%)	*-*	-	-
Missing	8	2	1	2	3	-	-	-
AFP	-	-	18.6 ± 7.4	41.8 ± 11.8	834.5 ± 156.4	19.24 ^a^	<0.0001	<0.0001
<20 ng/mL	153	NA	56 (91.8%)	55 (83.33%)	42 (60.9%)	19.71 ^b^	<0.0001	*-*
20–400 ng/mL	16	NA	5 (8.2%)	10 (15.15%)	1 (1.4%)	8.45 ^b^	0.015	-
>400 ng/mL	27	NA	0	1 (1.52%)	26 (37.7%)	51.2 ^b^	NA	-
Missing	5	NA	1	1	3	-	-	-
Hemoglobin (g/dL)	-	13.4 ± 0.3	12.9 ± 0.2	11.5 ± 0.22	12.8 ± 0.2	12.37 ^c^	<0.0001	<0.0001 ^e^
WBCs (×10^9^/L)	-	7.63 ± 0.42	4.75 ± 0.19	4.19 ± 0.09	4.614 ± 0.1	61.1 ^a^	<0.0001	0.015 ^d^
RBCs (×10^3^/mm^3^)	-	4.77 ± 0.274	12.9 ± 0.226	11.491 ± 0.22	12.79 ± 0.23	106.9 ^a^	<0.0001	<0.0001 ^d^
Platelets (×10^9^/L)	-	301 ± 12.4	220 ± 13.7	130 ± 7.6	141 ± 9.1	84.6 ^a^	<0.0001	0.481 ^d (N.S)^

Data are expressed as mean ± SEM; statistical significance is considered as *p-*value ≤ 0.05. Statistical analysis was performed using (^a^) Kruskal–Wallis, (^b^) Chi–square, (^c^) ANOVA, (^d^) Dunn’s posthoc, (^e^) Tukey HSD posthoc tests. NA: not applicable, as the number of participants in one or more groups (*n* = 0), N.S: Statistically non-significant.

**Table 2 cancers-14-03036-t002:** Non-invasive indices for determination of fibrosis and cirrhosis degrees.

		Groups			Statistics	
Parameter	No. of Participants	HCV Non-Cirrhotic	HCV Cirrhotic	HCV-HCC	χ^2 (a)^	*p-*Value
APRI score	213	0.424 ± 0.051	0.992 ± 0.13	01.365 ± 0.202	102.65	<0.0001
<0.5	113	36 (76.6%)	20 (33.9%)	16 (24.6%)	73.747	<0.0001
0.5–1.5	74	10 (21.3%)	29 (49.2%)	34 (52.3%)	37.409	<0.0001
>1.5	26	1 (2.1%)	10 (16.9%)	15 (23.1%)	20.165	<0.0001
Missing	32	15	8	7	-	-
FIB-4 score	192	1.684 ± 0.204	3.767 ± 0.399	4.534 ± 0.476	86.193	<0.0001
<1.45	67	23 (62.2%)	3 (6%)	7 (11.1%)	85.387	<0.0001
1.45–3.27	71	10 (27%)	29 (58%)	25 (39.7%)	18.686	<0.0001
>3.27	54	4 (10.8%)	18 (36%)	31 (49.2%)	34.641	<0.0001
Missing	53	25	17	9	-	-
AAR	238	1.14 ± 0.041	1.17 ± 0.043	1.25 ± 0.047	9.73	0.001
≤1	54	15 (24.6%)	17 (26.2%)	11 (15.7%)	2.805	0.423
>1	184	46 (75.4%)	48 (73.8%)	59 (84.3%)	-	-
Missing	7	1	2	2	-	-

Data are expressed as mean ± SEM; statistical significance is considered as *p-*value ≤ 0.05. ^a^: Statistical analysis was performed using the Chi-square test.

**Table 3 cancers-14-03036-t003:** Characteristics and staging of HCC patients.

Parameter	No. of Participants		HCV Cirrhotic	HCV-HCC
CTP score	A	100	46 (69.7%)	54 (78.3%)
	B	30	16 (24.2%)	14 (20.3%)
	C	5	4 (6.1%)	1 (1.4%)
	Missing	4	1	3
Ascites	Absent	117	20 (29.8%)	35 (48.6%)
	Present	31	15 (22.4%)	16 (22.2%)
	Unknown	53	32 (47.8%)	21 (29.2%)
Focal lesions number	Single	-	-	46 (63.9%)
	Multiple	-	-	26 (36.1%)
Focal lesions size (Measured by CT)	Tumor ≤ 3 cm	-	-	22 (40.7%)
	Tumor > 3 cm	-	-	32 (59.2%)
	Missing	-	-	18
Performance status (PS)	PS = 0	-	-	35 (48.7%)
	PS = 1–2	-	-	31 (43%)
	PS > 2	-	-	6 (8.3%)
BCLC staging system	0	-	-	10 (17.9%)
	A	-	-	36 (64.2%)
	B	-	-	10 (17.9%)
	Missing	-	-	16

**Table 4 cancers-14-03036-t004:** Mean of ∆Ct of target miRNAs among the studied groups.

Target	Groups				Statistics		
	Control Mean ∆Ct (*n* = 41)	HCV Non- cirrhotic Mean ∆Ct (*n* = 44)	HCV Cirrhotic Mean ∆Ct (*n* = 51)	HCV-HCC Mean ∆Ct (*n* = 52)	Test Statistics ^(a)^	*p*-Value (Cont. vs. HCV and HCC) ^(b)^	*p*-value (HCV vs. HCC) ^(b)^
miR-142	3.847 ± 0.42	−0.816 ± 0.47	−1.58 ± 0.5	−2.25 ± 0.32	74.022	<0.0001	0.625 ^(N.S.)^
miR-150	−2.402 ± 0.55	−7.42 ± 0.35	−7.37 ± 0.35	−8.49 ± 0.33	64.834	<0.0001	0.109 ^(N.S.)^
miR-183	4.21 ± 0.401	−0.254 ± 0.29	−1.24 ± 0.303	−1.22 ± 0.32	80.109	<0.0001	0.771 ^(N.S.)^
miR-199a	2.13 ± 0.49	−2.62 ± 0.35	−3.63 ± 0.32	−4.213 ± 0.26	81.468	<0.0001	0.429 ^(N.S.)^
miR-215	4.81 ± 0.45	0.217 ± 0.29	0.148 ± 0.27	−0.374 ± 0.21	76.745	<0.0001	0.264 ^(N.S.)^
mir-217	4.684 ± 0.48	−0.54 ± 0.395	−0.759 ± 0.33	−0.358 ± 0.29	72.422	<0.0001	0.521 ^(N.S.)^
miR-224	3.469 ± 0.46	−0.012 ± 0.41	−0.737 ± 0.3	−1.294 ± 0.25	60.309	<0.0001	0.34 ^(N.S.)^
miR-424	2.401 ± 0.45	−2.79 ± 0.37	−3.663 ± 0.36	−4.796 ± 0.303	90.225	<0.0001	0.05
miR-3607	3.358 ± 0.4	0.77 ± 0.25	0.93 ± 0.17	0.317 ± 0.21	46.934	<0.0001	0.479 ^(N.S.)^

Data are expressed as mean ± SEM; statistical significance is considered *p-*value ≤ 0.05. (N.S.) indicates the absence of statistical significance. Statistical analysis was performed using (^a^) Kruskal–Wallis or (^b^) Dunn’s posthoc test.

**Table 5 cancers-14-03036-t005:** Fold change mean rank of target miRNAs among the studied groups.

Target	Groups				Statistics	
	Control Fold Change Mean Rank (*n* = 41)	HCV Non-Cirrhotic Fold Change Mean Rank (*n* = 44)	HCV Cirrhotic Fold Change Mean Rank (*n* = 51)	HCV-HCC Fold Change Mean Rank (*n* = 52)	Test Statistics ^(a)^	*p*-Value
miR-142	31.634	98.659	115.098	120.346	74.0224	<0.0001
miR-150	35.976	104.636	104.235	122.519	64.3728	<0.0001
miR-183	28.879	99.705	119.971	116.856	80.1087	<0.0001
miR-199a	30.39	92.886	116.431	124.904	81.4677	<0.0001
miR-215	31.488	104.989	109.069	121.019	72.812	<0.0001
mir-217	31.39	111.875	119.039	105.49	72.4221	<0.0001
miR-224	38.146	98.091	110.235	120.462	60.2673	<0.0001
miR-424	27.049	95.568	111.549	130.058	90.2253	<0.0001
miR-3607	44.634	108.92	99.4314	116.779	46.9342	<0.0001

Statistical significance is considered *p-*value ≤ 0.05. Statistical analysis was performed using (^a^) Kruskal–Wallis. The mean rank is the average of the ranks for all observations within each study group calculated using SPSS statistical software.

**Table 6 cancers-14-03036-t006:** Correlation between the target miRNAs in the study groups and other clinicopathological factors.

Factor	Statistics	miR-142 2^−∆∆Ct^	miR-150 2^−∆∆Ct^	miR-183 2^−∆∆Ct^	miR-199a 2^−∆∆Ct^	miR-215 2^−∆∆Ct^	miR-217 2^−∆∆Ct^	miR-224 2^−∆∆Ct^	miR-424 2^−∆∆Ct^	miR-3607 2^−∆∆Ct^
miR-142 2^−∆∆Ct^	rho	1.000	0.619 **	0.403 **	0.624 **	0.371 **	0.350 **	0.579 **	0.651 **	0.359 **
	*p*-value	-	<0.0001	<0.0001	<0.0001	<0.0001	<0.0001	<0.0001	<0.0001	<0.0001
miR-150 2^−∆∆Ct^	rho	0.619 **	1.000	0.378 **	0.656 **	0.409 **	0.315 **	0.767 **	0.774 **	0.420 **
	*p*-value	<0.0001	-	<0.0001	<0.0001	<0.0001	<0.0001	<0.0001	<0.0001	<0.0001
miR-183 2^−∆∆Ct^	rho	0.403 **	0.378 **	1.000	0.617 **	0.707 **	0.725 **	0.439 **	0.495 **	0.497 **
	*p*-value	<0.0001	<0.0001	-	<0.0001	<0.0001	<0.0001	<0.0001	<0.0001	<0.0001
miR-199a 2^−∆∆Ct^	rho	0.624 **	0.656 **	0.617 **	1.000	0.659 **	0.541 **	0.727 **	0.828 **	0.581 **
	*p*-value	<0.0001	<0.0001	<0.0001	-	<0.0001	<0.0001	<0.0001	<0.0001	<0.0001
miR-215 2^−∆∆Ct^	rho	0.371 **	0.409 **	0.707 **	0.659 **	1.000	0.681 **	0.527 **	0.582 **	0.621 **
	*p*-value	<0.0001	<0.0001	<0.0001	<0.0001	-	<0.0001	<0.0001	<0.0001	<0.0001
miR-217 2^−∆∆Ct^	rho	0.350 **	0.315 **	0.725 **	0.541 **	0.681 **	1.000	0.339 **	0.441 **	0.503 **
	*p*-value	<0.0001	<0.0001	<0.0001	<0.0001	<0.0001	-	<0.0001	<0.0001	<0.0001
miR-224 2^−∆∆Ct^	rho	0.579 **	0.767 **	0.439 **	0.727 **	0.527 **	0.339 **	1.000	0.769 **	0.569 **
	*p*-value	<0.0001	<0.0001	<0.0001	<0.0001	<0.0001	<0.0001	-	<0.0001	<0.0001
miR-424 2^−∆∆Ct^	rho	0.651 **	0.774 **	0.495 **	0.828 **	0.582 **	0.441 **	0.769 **	1.000	0.545 **
	*p*-value	<0.0001	<0.0001	<0.0001	<0.0001	<0.0001	<0.0001	<0.0001	-	<0.0001
miR-3607 2^−∆∆Ct^	rho	0.359 **	0.420 **	0.497 **	0.581 **	0.621 **	0.503 **	0.569 **	0.545 **	1.000
	*p*-value	<0.0001	<0.0001	<0.0001	<0.0001	<0.0001	<0.0001	<0.0001	<0.0001	
Age	rho	0.021	0.049	0.206 **	0.160 *	0.170 *	0.081	0.082	0.127	−0.047
	*p*-value	0.781	0.527	0.007	0.037	0.026	0.291	0.285	0.098	0.538
Gender	rho	0.153 *	0.103	0.168 *	0.074	0.084	0.251 **	0.046	0.056	0.055
	*p*-value	0.036	0.159	0.021	0.315	0.252	0.001	0.530	0.442	0.450
Cirrhosis	rho	0.472 **	0.385 **	0.485 **	0.532 **	0.418 **	0.360 **	0.424 **	0.535 **	0.278 **
	*p*-value	<0.0001	<0.0001	<0.0001	<0.0001	<0.0001	<0.0001	<0.0001	<0.000	<0.0001
ALT	rho	0.452 **	0.402 **	0.404 **	0.481 **	0.422 **	0.322 **	0.483 **	0.511 **	0.268 **
	*p*-value	<0.0001	<0.0001	<0.0001	<0.0001	<0.0001	<0.0001	<0.0001	<0.0001	<0.0001
AST	rho	0.379 **	0.299 **	0.323 **	0.402 **	0.356 **	0.244 **	0.387 **	0.416 **	0.202 **
	*p*-value	<0.0001	<0.0001	<0.0001	<0.0001	<0.0001	<0.0001	<0.0001	<0.0001	<0.0001
AFP	rho	0.131	−0.067	0.081	0.170 *	0.082	−0.046	0.082	0.127	0.090
	*p*-value	0.117	0.420	0.332	0.040	0.324	0.583	0.328	0.128	0.281
T. Bil	rho	−0.064	−0.219 **	−0.008	−0.006	−0.005	−0.139	−0.057	−0.059	−0.139
	*p*-value	0.392	0.003	0.915	0.930	0.945	0.060	0.443	0.427	0.059
D. Bil	rho	0.293 **	0.179 *	0.423 **	0.364 **	0.423 **	0.389 **	0.312 **	0.353 **	0.284 **
	*p*-value	<0.0001	0.015	<0.0001	<0.0001	<0.0001	<0.0001	<0.0001	<0.0001	<0.0001
ALB	rho	−0.211 **	−0.071	−0.229 **	−0.258 **	−0.266 **	−0.159 *	−0.195 **	−0.265 **	−0.132
	*p*-value	0.004	0.339	0.002	<0.0001	<0.0001	0.031	0.008	<0.0001	0.074
CTP score	rho	−0.037	−0.259 **	0.154	0.055	0.123	0.231 *	−0.057	0.045	0.214 *
	*p*-value	0.715	0.009	0.124	0.587	0.219	0.020	0.570	0.658	0.032
BCLC	rho	−0.295	−0.110	−0.035	0.136	0.168	−0.042	−0.210	−0.115	−0.056
	*p*-value	0.065	0.500	0.829	0.404	0.301	0.798	0.194	0.481	0.730

Association was determined using Spearman’s correlation. Rho: Spearman’s rank correlation coefficient. * Correlation is significant as *p-*value ≤ 0.05 (two-tailed), ** Correlation is significant as *p-*value ≤ 0.01 (two-tailed).

**Table 7 cancers-14-03036-t007:** Diagnostic accuracy of single and combined miRNAs between HCC relative to healthy control and non-HCC individuals.

miRNA	AUC	95% CI	Cut-Off	Sensitivity	Specificity	PPV	NPV	Accuracy	SE	*p*-Value
HCC vs. Control										
miR-424	0.993	0.98–1	9.05	100	90.24	92.86	100	95.7	0.005	<0.0001
miR-199a	0.968	0.93–1	18.22	92.31	95.12	96	90.7	93.55	0.02	<0.0001
miR-142	0.972	0.95–0.99	10.80	92.31	90.24	92.31	90.24	91.4	0.014	<0.0001
miR-215	0.958	0.92–0.994	13.89	92.31	90.24	92.31	90.24	91.4	0.019	<0.0001
miR-224	0.921	0.87–0.97	9.60	80.77	87.8	89.36	78.26	83.87	0.027	<0.0001
miR-150	0.928	0.88–0.98	10.36	88.46	82.93	86.8	85	86.02	0.024	<0.0001
miR-3607	0.868	0.79–0.95	3.96	82.7	80.49	84.31	78.57	81.72	0.041	<0.0001
miR-183	0.957	0.92–0.998	9.14	94.23	90.24	92.45	92.5	92.47	0.021	<0.0001
miR-217	0.933	0.88–0.98	9.54	86.54	85.37	88.24	83.33	86.02	0.026	<0.0001
Combined panel ^a^	0.969	0.928–1		100	95.12	96.3	100	97.85	0.021	<0.0001
HCV vs. Control										
miR-424	0.940	0.9–0.98	7.173	85.26	85.37	93.1	71.43	85.29	0.019	<0.0001
miR-199a	0.919	0.86–0.97	8.418	84.21	85.37	93.02	70	84.56	0.028	<0.0001
miR-142	0.903	0.85–0.96	7.165	86.32	85.37	93.18	72.92	86.03	0.026	<0.0001
miR-215	0.913	0.86–0.96	6.196	86.32	80.49	91.11	71.74	84.56	0.06	<0.0001
miR-224	0.863	0.8–0.93	5.729	80	78.05	89.41	62.74	79.41	0.034	<0.0001
miR-150	0.882	0.83–0.94	6.966	82.11	73.17	87.64	63.83	79.41	0.029	<0.0001
miR-3607	0.824	0.74–0.91	3.271	81.05	75.61	88.5	63.27	79.41	0.044	<0.0001
miR-183	0.941	0.9–0.98	6.266	91.58	87.8	94.57	81.82	90.44	0.022	<0.0001
miR-217	0.927	0.88–0.97	8.496	88.42	85.37	93.33	76.1	87.5	0.023	<0.0001
Combined panel ^a^	0.871	0.8–0.941		90.53	85.37	93.48	79.55	88.97	0.036	<0.0001
HCC vs. non-HCC										
miR-424	0.761	0.7–0.83	27.943	80.77	60.29	43.75	89.13	65.96	0.035	<0.0001
miR-199a	0.724	0.65–0.8	28.769	78.85	58.09	41.84	87.78	63.83	0.037	<0.0001
miR-142	0.69	0.61–0.77	21.811	76.92	58.09	41.24	86.81	63.3	0.039	<0.0001
miR-215	0.695	0.62–0.77	22.138	73.08	55.15	38.38	84.27	60.11	0.039	<0.0001
miR-224	0.691	0.61–0.77	10.321	73.08	54.41	38	84.1	59.57	0.04	<0.0001
miR-150	0.706	0.63–0.79	23.534	71.15	54.41	37.38	83.15	59.04	0.041	<0.0001
miR-3607	0.664	0.58–0.75	5.797	71.15	52.94	36.63	82.76	57.98	0.042	0.001
miR-183	0.664	0.59–0.74	18.872	61.54	59.56	36.78	80.2	60.11	0.041	<0.0001
Combined panel ^b^	0.68	0.593–0.767		80.77	61.03	44.21	89.25	66.49	0.044	<0.0001
HCC vs. non-HCC (SVR group)										
miR-424	0.8	0.72–0.89	24.165	79.17	62.5	32.76	92.86	65.63	0.044	<0.0001
miR-199a	0.74	0.65–0.83	28.769	83.33	64.42	35.1	94.37	67.97	0.045	<0.0001
miR-142	0.794	0.71–0.88	38.561	75	70.19	36.73	92.41	71.1	0.042	<0.0001
miR-215	0.742	0.65–0.83	24.654	75	64.42	32.73	91.78	66.41	0.045	<0.0001
miR-224	0.703	0.6–0.8	11.042	70.83	61.54	29.82	90.14	63.28	0.051	0.002
miR-150	0.735	0.63–0.84	23.534	75	59.62	30	91.18	62.5	0.053	<0.0001
miR-3607	0.763	0.67–0.86	8.695	75	66.35	33.96	92	67.97	0.05	<0.0001
miR-183	0.728	0.63–0.82	18.872	62.5	65.38	29.41	88.31	64.84	0.048	0.001
miR-217	0.673	0.57–0.78	24.745	75	57.7	29.03	90.91	60.94	0.052	0.008
Combined panel ^a^	0.683	0.562–0.804		83.33	63.73	35.09	94.2	67.46	0.062	0.005
HCC vs. non-HCC (Naïve group)										
miR-424	0.835	0.76–0.91	29.098	85.71	75.34	57.14	93.22	78.22	0.039	<0.0001
miR-199a	0.821	0.74–0.9	22.67	89.29	68.5	52.1	94.34	74.26	0.041	<0.0001
miR-142	0.737	0.64–0.83	12.933	85.71	64.38	48	92.16	70.3	0.049	<0.0001
miR-215	0.772	0.68–0.86	17.087	85.71	63.01	47.06	92	69.31	0.046	<0.0001
miR-224	0.78	0.7–0.87	9.76	82.14	61.64	45.1	90	67.33	0.046	<0.0001
miR-150	0.791	0.7–0.88	11.152	82.14	60.27	44.23	89.8	66.34	0.046	<0.0001
miR-3607	0.695	0.59–0.8	5.0867	71.43	61.64	41.67	84.91	64.36	0.052	0.002
miR-183	0.748	0.65–0.84	14.496	85.71	64.38	48	92.16	70.3	0.049	<0.0001
miR-217	0.677	0.58–0.78	13.518	75	60.27	42	86.27	64.36	0.052	0.006
Combined panel ^c^	0.809	0.718–0.901		89.29	72.6	55.56	94.64	77.23	0.047	<0.0001
HCC vs. HCV (cirrhotic and non-cirrhotic)										
miR-424	0.662	0.57–0.75	79.387	63.46	57.9	45.21	74.32	59.86	0.046	0.001
miR-199a	0.618	0.53–0.71	38.725	63.46	50.52	41.25	71.64	55.1	0.047	0.018
miR-150	0.61	0.51–0.71	45.166	55.77	50.53	38.16	67.61	52.38	0.05	0.028
AFP	0.688	0.60–0.77	6.25	62.32	64.57	48.86	75.93	63.78	0.043	<0.0001
Combined panel ^d^	0.567	0.469–0.644		61.54	56.84	43.84	72.97	58.5	0.05	0.182 ^(ns)^
Combined panel ^d^ + AFP	0.61	0.512–0.707		62.75	56.84	43.84	73.97	58.9	0.05	0.03
HCC vs. HCV (cirrhotic and non-cirrhotic) (SVR)										
miR-424	0.679	0.55–0.81	79.387	66.67	60.32	39.02	82.61	62.07	0.065	0.01
miR-142	0.665	0.54–0.79	112.42	66.67	69.84	45.71	84.62	68.97	0.063	0.018
miR-3607	0.672	0.54–0.8	9.325	70.83	53.97	36.96	82.93	58.62	0.067	0.014
AFP	0.615	0.5–0.73	6.05	51.43	60.67	33.96	76.06	58.065	0.059	0.047
Combined panel ^d^	0.653	0.514–0.792		70.83	61.9	41.46	84.78	64.37	0.071	0.028
Combined panel ^d^ + AFP	0.676	0.542–0.809		70.83	73.02	50	86.79	72.41	0.068	0.012

Statistical significance is considered *p-*value ≤ 0.05. The combined panel was calculated with: (^a^) 5 out of 9 significantly altered candidate miRNAs, (^b^) 5 out of 8 significantly altered candidate miRNAs, (^c^) 6 out of 9 significantly altered candidate miRNAs, and (^d^) 2 out of 3 significantly altered candidate miRNAs. ns: Statistically non-significant. NPV: negative predictive value, PPV: positive predictive value.

**Table 8 cancers-14-03036-t008:** Using logistic regression analysis, predict the probability of single and combined miRNAs in HCC patients relative to healthy controls, HCV, and non-HCC individuals.

miRNA	Normalized LR Weights	Cut-off	AUC	Specificity	Sensitivity	Accuracy
HCC vs. Control						
miR-150	0.205523	-	-	-	-	-
miR-199a	0.145376	-	-	-	-	-
miR-215	0.145271	-	-	-	-	-
miR-424	0.138455	-	-	-	-	-
miR-224	0.127198	-	-	-	-	-
miR-217	0.097305	-	-	-	-	-
miR-183	0.074394	-	-	-	-	-
miR-142	0.064225	-	-	-	-	-
miR-3607	0.002254	-	-	-	-	-
Combined panel		0.121	0.95	95.12	94.23	94.62
HCV vs. Control						
miR-424	0.182482	-	-	-	-	-
miR-142	0.155125	-	-	-	-	-
miR-215	0.142391	-	-	-	-	-
miR-224	0.142301	-	-	-	-	-
miR-150	0.132367	-	-	-	-	-
miR-183	0.074106		-	-	-	-
miR-199a	0.070252	-	-	-	-	-
miR-217	0.062827	-	-	-	-	-
miR-3607	−0.03815	-	-	-	-	-
Combined panel		0.101	0.91	90.24	91.58	91.18
HCC vs. non-HCC						
miR-150	0.280744	-	-	-	-	-
miR-424	0.164503	-	-	-	-	-
miR-199a	0.124309	-	-	-	-	-
miR-142	0.10468	-	-	-	-	-
miR-183	0.094862	-	-	-	-	-
miR-215	0.088024	-	-	-	-	-
miR-224	0.058672	-	-	-	-	-
miR-3607	0.048511	-	-	-	-	-
miR-217	−0.03569	-	-	-	-	-
Combined panel		0.111	0.65	65.44	65.38	65.43
HCC vs. HCV						
miR-150	0.281315	-		-	-	-
miR-424	0.190728	-	-	-	-	-
miR-3607	0.122029	-	-	-	-	-
miR-183	0.098993	-	-	-	-	-
miR-224	0.092565	-	-		-	-
miR-199a	0.065576	-	-	-	-	-
miR-215	0.043295	-	-	-	-	-
miR-142	0.033518	-	-	-	-	-
miR-217	−0.07198	-	-	-	-	-
Combined panel		0.182	0.62	62.11	61.54	61.9

The combined panels were calculated using the nine candidate miRNAs.

**Table 9 cancers-14-03036-t009:** Common pathways identified in DE-miRNAs.

miRNA	Related Pathways
miR-142	Cell migration, neoplasm metastasis, colony formation, cell growth, neoplasms, apoptosis, cell cycle, cell proliferation
miR-150	cell migration, neoplasm metastasis, colony formation, vascularization, breast cancer, cell growth, neoplasms, apoptosis, cell cycle
miR-183	Breast cancer related
miR-199a	Melanoma, breast cancer related, liver cancer, cell migration, renal cell carcinoma, neoplasm metastasis, cell proliferation
miR-215	Liver cancer, cell migration, cell proliferation, cell cycle arrest, G1/S transition checkpoint, colorectal cancer, hepatocellular carcinoma
miR-217	Cell migration, renal cell carcinoma, neoplasm metastasis, cell proliferation, hypertrophy, senescence, apoptosis, colony formation, lumen formation
miR-224	Breast cancer related
miR-3607	Liver cancer

## Data Availability

The datasets analyzed during the current study are available in the GEO-NCBI repository (https://www.ncbi.nlm.nih.gov/geo/query/acc.cgi?acc=GSE40744, accessed on 3 May 2021) and TCGA repository (https://www.cell.com/cell/fulltext/S0092-8674(17)30639-6 and https://www.cancer.gov/about-nci/organization/ccg/research/structural-genomics/tcga/studied-cancers/liver, accessed on 3 May 2021).

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
