# Peer review of "Prognostic MicroRNA Panel for HCV-Associated HCC: Integrating Computational Biology and Clinical Validation"

_cancers, 2022, doi:10.3390/cancers14133036_

Round 1

Reviewer 1 Report

Although the presentation of the manuscript is well, the idea of miRNAs as biomarkers of HCV-related HCC are extensively studied and thus the work is of no sufficient novelty. The combined panels also of mild or moderate discrimination between HCC vs non-HCC gps (Table 7) even after addition of AFP. My Comments are

1- The sample size is relatively low. How did the authors calculate the sample size for this study?

2- The AUCs of combined panels in table 7 are missed. It should be added to the table and the panels should be presented in the ROC figures.

3- The authors didn't describe how the combined panels were generated e.g., through logistic regression analysis? A full description of the method should be added.

4- A logistic regression analysis was not done to predict HCC among non-HCC groups. I think this will add much to display the predictive ability of the microRNAs.

5- In table 4, if the authors used Kruskal Wallis test, Dunn's multiple comparison test should be used for multiple comparisons, the use of Mann Whitney for multiple comparisons yields errors.

6- In table 5, the method for calculating the control fold mean rank should be identified in the table legend.

7- The authors should do some work in target gene analysis or interaction networks between the described miRNAs in relation to HCC. This will add a comprehensive view and break the descriptive discussion of previous literature.

8- Why 61% of HCC patients have almost normal AFP (< 20 ng/ml) in this study? The authors should explain in the discussion.

9- A paragraph containing the study limitations should be added to the end of the discussion.

10- The number of references should be reduced.

Author Response

Comments from reviewer 1:

Q.1- The sample size is relatively low. How did the authors calculate the sample size for this study?

A.1- Thank you for pointing this out. We agree with this comment. Therefore, we have included the low sample size as one of our study limitations, which could be found in line 824. However, the sample size determination was calculated using the following formula https://www.ncbi.nlm.nih.gov/pmc/articles/PMC3775042/ :

Sample size = [Z(1-α/2)2 x p (1- p)] / d2

Where Z(1-α/2) is the standard normal variate (at 5% type 1 error (P < 0.05)) = 1.96.

p = Expected proportion in population based on previous studies or pilot studies

d = Absolute error or precision  We have chosen margin of error of 5%

Sample size = [1.962 x 0.04(1-0.04)] / 0.052 = 59.007

Given that around 4% of the Egyptian population (age between 1-59) had active HCV infection and almost 6 million Egyptians in the age group of 15-59 were chronically infected with HCV, according to the Egyptian demographic health survey performed by the Egyptian ministry of health in 2015. Thus the calculated sample size per group was 60. Therefore, we included 62 HCV-noncirrhotic, 67 HCV-cirrhotic, and 72 HCV-HCC patients.

Q.2- The AUCs of combined panels in table 7 are missed. It should be added to the table and the panels should be presented in the ROC figures.

A.2- Thanks for your valuable comment. Table 7 was modified by adding the combined panels' AUC and confidence interval values. In addition to figure 13 that was constructed showing the ROC curves corresponding to the integrated panels of the differentially expressed miRNAs among the study groups.

Q.3- The authors didn't describe how the combined panels were generated e.g., through logistic regression analysis? A full description of the method should be added.

A.3- We have incorporated the method by which the combined panels were generated in the manuscript, section 2.5 in the methodology, lines 219-226. It was described as the following: “Receiver operator characteristic (ROC) curve was constructed to identify the optimal cutoff value which maximizes the sum of sensitivity and specificity of each biomarker and area under the curve (AUC) was calculated. The performances of the individual targets and the combined panels were evaluated by comparing the predicted outcome values (probability of true positive and false positive) with the true outcome variables and calculating the sensitivity and specificity. The optimal cut-off value, sensitivity, and specificity were determined by calculating the Youden index”. The same methodology was previously applied in some research studies, including Moustafa et al., https://pubmed.ncbi.nlm.nih.gov/32165186/, and Lia et al., https://www.ncbi.nlm.nih.gov/pmc/articles/PMC6366967/.

Q.4- A logistic regression analysis was not done to predict HCC among non-HCC groups. I think this will add much to display the predictive ability of the microRNAs.

A.4- We have included logistic regression analysis in determining the predictive ability of the individual miRNAs and the combined panels in the methodology section 2.6, lines 228-237, and in the results section 3.7, lines 536-607 and the discussion was modified accordingly, lines 782-796.

Q.5- In table 4, if the authors used Kruskal Wallis test, Dunn's multiple comparison test should be used for multiple comparisons, the use of Mann Whitney for multiple comparisons yields errors.

A.5- Thank you for pointing this out. We have modified the calculations in table 4 based on Dunn’s post hoc test instead of Mann’s Whitney test.

Q.6- In table 5, the method for calculating the control fold mean rank should be identified in the table legend.

A.6- The authors agree with the reviewer's comment and have included the method of calculating the mean rank in table 5 legend. It is described as follows: “The mean rank is the average of the ranks for all observations within each study group, and it was calculated using SPSS statistical software.”

Q.7- The authors should do some work in target gene analysis or interaction networks between the described miRNAs in relation to HCC. This will add a comprehensive view and break the descriptive discussion of previous literature.

A.7- The authors are thankful for this reviewer’s suggestion, which has significantly enriched the manuscript. We have performed target gene analysis describing the interaction between the target miRNAs in relation to liver cancer generally and to HCC specifically. The methodology of the newly added analysis is titled Pathway Analysis section 2.7, lines 239-250. The results section includes interpretation of the pathway analysis section 3.8, lines 609-644. The discussion was modified based on the new results in lines 797-811.

Q.8- Why 61% of HCC patients have almost normal AFP (< 20 ng/ml) in this study? The authors should explain in the discussion.

A.8- This is an excellent point raised by the reviewer, and the authors have had the same question. In this regard, it was previously mentioned in several research studies that although AFP is considered the most extensively chosen circulatory biomarker for HCC, it is characterized by inconvenient sensitivity and specificity in the determination of HCC lesions even at low-level cutoffs (10-20 ng/mL) https://www.journal-of-hepatology.eu/article/S0168-8278(00)00053-2/fulltext?refuid=S0168-8278(11)00441-7&refissn=0168-8278. In addition, it has 25% sensitivity in the identification of small-sized lesions (< 3 cm), and the sensitivity could reach 50% for focal lesions larger than 3 cm https://www.ncbi.nlm.nih.gov/pmc/articles/PMC2811792/. Therefore, in the current study, we investigated the diagnostic potential of serum miRNAs alone or in conjunction with AFP to improve the diagnostic accuracy of AFP, especially in the early disease stages. Thus, HCC patients enrolled in the study were classified into BCLC stages 0, A, and B (i.e., very early, early, and intermediate stages). And according to the statistics reported in table 3, 64% of HCC patients have a single hepatic focal lesion, and 40.7% of HCC patients have a focal lesion size < 3cm. Moreover, our ROC analysis showed that AFP had a sensitivity of 62.3% and a specificity of 64.5% in diagnosing HCC patients enrolled in the study. Therefore, we believe that those findings were compatible with the normal or low AFP levels in 61% of the HCC study group. This fact was also supported by previous studies https://www.frontiersin.org/articles/10.3389/fonc.2020.01316/full , https://www.ncbi.nlm.nih.gov/pmc/articles/PMC2023919/ , https://www.ncbi.nlm.nih.gov/pmc/articles/PMC5865641/ , https://www.mdpi.com/2077-0383/8/10/1736/htm . Accordingly, to address this specific point, we included the explanation of the normal AFP levels in HCC patients in discussion lines 663-669.

Q.9- A paragraph containing the study limitations should be added to the end of the discussion.

A.9- Thanks for your valuable comment. Accordingly, we have included a paragraph explaining the study limitations at the end of the discussion lines 823-834.

Q.10- The number of references should be reduced.

A.10- Thanks for your comment. The cited references were reduced accordingly.

Reviewer 2 Report

In the present study, Dabbish and colleagues investigated the the role of a panel of miRNAs as biomarkers for the identification of patients with HCC. The study cohort included healthy subjects, patients with chronic HCV infection, patients with HCV-related liver cirrhosis and patients with HCC. The topic addressed is clinically relevant; though the availability of novel potent antivirals that grant high rates of sustained virologic response, patients with cirrhosis cured from HCV infection are still at risk of HCC development. Therefore, the identification of reliable biomarkers for the detection of HCC is an unmet need. 

In this regard, the major limitation of the manuscript is represented by study population; diagnostic accuracy analyses were performed comparing patients with HCC vs those without (irrespectively from the presence of liver cirrhosis), even in comparison with healthy subjects. The appropriate control population is made of patients with cirrhosis, that are those at risk of HCC development. Furthemore, the number of patients enrolled (67 HCV-related cirrhosis and 72 HCV-related HCC) is an additional major limitation of the study. 

Specific comments:
1) Materials and methods, page 4, lines 147-150. Diagnosis of HCV based on ultrasonography?? US can be useful for the detection of features of cirrhosis. Was the degree of fibrosis assessed only by means of non invasive test (APRI, FIB-4, etc)? Are liver stiffness data available?

2) Table 1. Please add a p value for the specific comparison cirrhotic and HCC groups. Furthemore, variables such as albumin, bilirubin, AFP, hemoglobin requires only 1 decimal value, while variables such as AST, ALT and platelets should be reported without decimals.

3) Lines 269-275. I cannot understand the rationale of assessing the value of non-invasive test for the prediction of significant fibrosis and advance fibrosis in patients with cirrhosis. Please explain, considering also the comment 1.

4) While I accept the investigation of miRNAs expression in different study groups (i.e. healthy subjects, chronic hepatitis, cirrhosis, HCC), from page 13, the result section need to be completely amended focusing on assessing the diagnostic accuracy of the miRNAs for the discrimination between patients with cirrhosis vs patients with HCC. Furthemore, while performing the new analysis, authors must take into account the potential confounding factors of demographic, biochemical, and clinical variables (for instance difference in age, gender, parameters of liver function, etc) between patients with HCC and those with cirrhosis.

5) Discussion and conclusions should be revised according to the new results obtained.

Author Response

Comments from reviewer 2:

Q.1- Materials and methods, page 4, lines 147-150. Diagnosis of HCV based on ultrasonography?? US can be useful for the detection of features of cirrhosis. Was the degree of fibrosis assessed only by means of non-invasive test (APRI, FIB-4, etc)? Are liver stiffness data available?

A.1-The authors are grateful for this reviewer comment. Diagnosis of “cirrhosis,” not “HCV,” was done using ultrasonography, which was mistakenly written. Moreover, the degree of fibrosis was primarily done using ultrasonography, while non-invasive testing was performed as an additional evaluation technique. This was accordingly modified in the manuscript lines 151-157. Unfortunately, measurement of liver stiffness isn’t usually performed in assessing HCC patients in hospitals, as the price of elastography is unaffordable for most patients. Only a few HCC patients enrolled in the study have their liver stiffness measurements. However, including only the samples of the patients who got their “Fibro-scan results” will dramatically decrease the overall sample size, which would be an additional study limitation.

Q.2- Table 1. Please add a p value for the specific comparison cirrhotic and HCC groups. Furthermore, variables such as albumin, bilirubin, AFP, hemoglobin requires only 1 decimal value, while variables such as AST, ALT and platelets should be reported without decimals.  

A.2- The authors are grateful for this reviewer’s comment, and a new row of the P values comparing the HCV cirrhotic group to the HCC group was added. The lab test variables were modified based on the reviewer’s suggestion.

Q.3- Lines 269-275. I cannot understand the rationale of assessing the value of non-invasive test for the prediction of significant fibrosis and advance fibrosis in patients with cirrhosis. Please explain, considering also the comment 1.

A.3- The reviewer has raised an important point here. The authors would like to clarify that in the current study, the use of non-invasive indices was done as an alternative to the missing liver stiffness data, as previously supported in literature https://www.ncbi.nlm.nih.gov/pmc/articles/PMC6686098/ ,https://journals.lww.com/jfmpc/Fulltext/2021/11000/APRI_and_FIB_4_performance_to_assess_liver.23.aspx. Accordingly, an explanation was added in the manuscript lines 151-157 and lines 340-341 based on this reviewer’s suggestion. Notably, the authors believe that APRI, FIB-4, and AAR can assess the degree of cirrhosis based on the following literature. Wai et al. concluded that APRI and FIB-4 have a moderate degree of accuracy in detecting patients with advanced fibrosis and cirrhosis https://pubmed.ncbi.nlm.nih.gov/12883497/. At the same time, Papadopoulos et al. reported that the FIB-4 score threshold of 1.63 could predict cirrhotic patients https://www.ncbi.nlm.nih.gov/pmc/articles/PMC6686098/. Furthermore, Yunihastut et al.showed that APRI score ≥ 1 and FIB-4 score ≥ 1.66 had a moderate performance with high specificity in diagnosing cirrhosis https://bmcinfectdis.biomedcentral.com/articles/10.1186/s12879-020-05069-5. Additionally, according to the meta-analysis performed by Lin et al., APRI score greater than 1.0 was able to predict cirrhosis with a sensitivity of 76% and a specificity of 72% https://pubmed.ncbi.nlm.nih.gov/21319189/. Intriguingly, in a study performed by Leuştean and his colleagues, APRI and FIB‑4 proved to be easy, quick, and inexpensive tools for screening HCV cirrhosis, with moderate diagnostic accuracy. FIB‑4 was useful for monitoring patients post-treatment after achieving sustained virological response https://www.spandidos-publications.com/10.3892/etm.2020.9531.

Q.4- While I accept the investigation of miRNAs expression in different study groups (i.e. healthy subjects, chronic hepatitis, cirrhosis, HCC), from page 13, the result section need to be completely amended focusing on assessing the diagnostic accuracy of the miRNAs for the discrimination between patients with cirrhosis vs patients with HCC. Furthermore, while performing the new analysis, authors must take into account the potential confounding factors of demographic, biochemical, and clinical variables (for instance difference in age, gender, parameters of liver function, etc) between patients with HCC and those with cirrhosis.

A.4- The authors would like to thank this reviewer for his valuable suggestion. A new paragraph explaining the demographic, biochemical, and clinical differences between cirrhotic and HCC groups was added in the results section, lines 299-309, in addition to assessing the correlation between the target miRNAs in relation to HCV-cirrhotic and HCC groups only. The results were included under section 3.5, lines 399-416, and supplementary table 3. However, in the ROC analysis, we have already compared HCV- cirrhotic and HCC groups (included in sections 3.6.5 and 3.6.6). And we acknowledge showing the complete range of results among the whole study groups.

Q.5- Discussion and conclusions should be revised according to the new results obtained.

A.5- Thanks for your comment. We agree with this and have incorporated the new findings throughout the results and discussion sections. In addition to the above comments, all spelling and grammatical errors have been corrected.

Round 2

Reviewer 1 Report

The authors have changed as required.

Author Response

Thank you very much. The authors appreciate your valuable comments.

Reviewer 2 Report

The authors made great effort in order to improve the manuscript; however, given the suboptimal performance of miRNAs for the discrimination between patients with HCC and those with HCV-related cirrhosis (i.e. patients at risk of HCC development, and for this reason those who require surveillance and therefore the target population for biomarkers application), authors should soften the tone concerning the applicability of miRNAs as HCC biomarkers (at least at current stage of evidence). Accordingly, I suggest to amend the conclusions of the mansucript and of the abstract.

Author Response

The authors agree with the reviewer's comment. Accordingly, the abstract and conclusion sections have been amended.